

**Permafrost Stability Mapping on the Tibetan Plateau by Integrating**
**Time-series InSAR and Random Forest Method**
Fumeng Zhao[1], Wenping Gong[1]*, Tianhe Ren[1], Jun Chen[2,3], Huiming Tang[1], and Tianzheng Li[1]
[1]Faculty of Engineering, China University of Geosciences, Wuhan, Hubei 430074, China
[2]School of Automation, China University of Geosciences, Wuhan, Hubei 430074, China
[3]Hubei Key Laboratory of Advanced Control and Intelligent Automation for Complex Systems, Wuhan, Hubei 430074, China
*Correspondence to*: Wenping Gong (wenppinggong@cug.edu.cn)
**Abstract:** Ground deformation is an important index for evaluating the stability and degradation of the permafrost. Due to
limited accessibility, in-situ measurement of the ground deformation of permafrost area on the Tibetan Plateau is a challenge.
Thus, the technique of time-series Interferometric Synthetic Aperture Radar (InSAR) is often adopted for measuring the ground
deformation of the permafrost area, the effectiveness of which is however degraded in the areas with geometric distortions in
Synthetic Aperture Radar (SAR) images. In this study, a method that integrates InSAR and random forest method is proposed for
an improved permafrost stability mapping on the Tibetan Plateau; and, to demonstrate the application of the proposed method,
the permafrost stability mapping in a small area located in the central region of the Tibetan Plateau is studied. First, the ground
deformation in the concerned area is studied with InSAR, in which 67 Sentinel-1 scenes taken in the period from 2014 to 2020
are collected and analyzed. Second, the relationship between the environmental factors (i.e., topography, land cover, land surface
temperature, and distance-to-road) and the permafrost stability is mapped with the random forest method, based on the high-
quality data extracted from initial InSAR analysis. Third, the permafrost stability in the areas where the visibility of SAR images
is poor or the InSAR analysis results are not available is mapped with the trained random forest model. Comparative analyses
demonstrate that the integration of InSAR and random forest method yields a more effective permafrost stability mapping,
compared to the sole application of InSAR analysis.
**Keywords:** Permafrost Stability; InSAR; Random Forest Method; Tibetan Plateau.
**1. Introduction**
Under the influences of global warming and human activities, the mountain ecosystem and cryosphere system have changed
significantly, especially those at high altitudes and high latitudes (Chapin et al., 2005; Nicholas et al., 2015; Biskaborn et al.,
2019; Luo et al., 2019). As the third pole of the Earth, the Tibetan Plateau is sensitive to climate warming. The warming rate in
this plateau is about twice as high as the global warming rate over the past 40 years (Cheng et al., 2019). As a result, the
permafrost on the Tibetan Plateau has been degraded drastically, manifested in shrinking of permafrost boundaries, change of
permafrost types, increase of the thickness of active layer, emergence of thermokarst lakes, and even soil desertification (Liu et
al., 2013; Daout et al., 2017; Biskaborn et al., 2019; Huang et al., 2020; Turetsky et al., 2020; Lu et al., 2020). The degradation
of the permafrost will have negative impacts on the engineering facilities, ecosystem functions, and hydrogeological processes
on the Tibetan Plateau (Bense et al., 2009; Ran et al., 2021). It is worthwhile noting that with the increase of the thickness of
active layer (caused by the increase of the temperature), the organic carbon stored in the permafrost will be released into the
atmosphere, which could further amplify the regional and global climate warming (Biskaborn et al., 2019). Hence, it is
particularly important to monitor the deformation and stability of the permafrost on the Tibetan Plateau.



The permafrost stability is often evaluated based on the mean annual ground temperature (Ran et al., 2021), even though it can
be influenced by various factors. It should be noted that the ground deformation is fairly sensitive to the seasonal variation of the
ground temperature and the thickness of the active layer (Liu et al., 2013; Lu et al., 2019). The thickness of the active layer is
defined as the maximum annual thaw depth of the soil over permafrost, which is often taken as an index for characterizing the
permafrost stability (Schaefer et al., 2015). Climate warming tends to increase the amplitude of the ground temperature and that
of the ground moisture in each freeze-thaw cycle, which will then lead to an increase in the thickness of active layer and the
spatial and interannual variations of the quantity of water (or ice) stored in the active layer (Daout et al., 2017; Cheng et al.,
2019). The thickness of the active layer can be measured directly with grid probing, thaw tube, and ground penetrating radar
(Smith et al., 2009; Liu et al., 2013). Although these measurements are of high quality, they are sparse and the measurement
accuracy is site-specific (Daout et al., 2017; Widhalm et al., 2017). Indeed, similar problems exist in the monitoring of the
ground temperature. With the aid of the analytical model that is based on the heat conduction equation and the environmental
conditions (Shiklomanov et al., 2002), the thickness of the active layer monitored from the point measurement could be extended
to that at a regional scale. A potential limitation of this interpolation is too that many environmental factors are involved and the
determination of these factors can be a challenge (Widhalm et al., 2017).
As mentioned above, the ground deformation in a permafrost area can be sensitive to the ground temperature variation and the
active layer thickness; thus, the permafrost stability can be captured through the monitoring of the ground deformation. During
the past few decades, remote sensing techniques have become an indispensable tool for monitoring the ground deformation and
evaluating the permafrost stability in permafrost areas, owing to their wide coverage and independence of ground measurements
(Short et al., 2014; Widhalm et al., 2017; Wang et al., 2018b; Anderson et al., 2019; Gao et al., 2020). For example, the thickness
of the active layer in the permafrost area may be estimated based on the monitored ground deformations (Liu et al., 2013).
Among the various remote sensing techniques, Interferometric Synthetic Aperture Radar (InSAR), the effectiveness of which is
not affected by the weather condition, is quite popular because of its high accuracy in monitoring the small ground deformation
(Barnhart et al., 2018; Bekaert et al., 2020). The effectiveness of differential InSAR in monitoring the ground deformation and
the permafrost stability is, however, degraded by the spatial decoherence and atmospheric delay induced in the processing of
SAR images. Hence, the technique of time-series InSAR, such as persistent scatterer InSAR (PS-InSAR) and small baseline
subset-InSAR (SBAS-InSAR), has been developed recently and applied to monitoring the ground deformation and the
permafrost stability on the Tibetan Plateau (Schaefer et al., 2015; Daout et al., 2017; Lu et al., 2019). Indeed, apart from the
permafrost stability, the ground deformation monitored by the time-series InSAR can also be applied to the evaluation of slope
stability (Akbarimehr et al., 2013; Bar and Dixon 2021) and land subsidence (Motagh et al., 2007; Chaussard et al., 2014). For
example, the large ground deformation is often taken as evidence of slope failures or landslides. The effectiveness of the time-
series InSAR is, however, degraded in mountainous terrains, due to the geometric distortions (i.e., foreshortening, layover, and
shadow regions) in input SAR images (Colesanti and Wasowski 2006). In addition, the ground deformation in areas with dense
vegetation and water covering may not be monitored, due to the spatial decoherence induced in the processing of SAR images. In
other words, the ground deformation points detected by the time-series InSAR may not cover the entire study area.
In the field of landslide susceptibility mapping, the historical landslide information in a region is often collected and adopted for
training the relationship between environmental factors and landslide occurrence, and the trained relationship is then applied to
predict the probability of landslide occurrence in other regions with similar environmental conditions (Guzzetti et al., 2005;
Gemitzi et al., 2011). Inspired by the concept of landslide susceptibility mapping, a method that integrates the time-series InSAR
and machine learning method is proposed in this paper for an improved permafrost stability mapping on the Tibetan Plateau. The
integrated method could take the advantage of the effectiveness of time-series InSAR (in monitoring the ground deformation in



the areas with good visibility of input SAR images) and that of the machine learning method (in mapping the relationship
between the environmental factors and the ground deformation). With the aid of the trained relationship between the
environmental factors and the permafrost stability, the permafrost stability in the areas where the visibility of SAR images is
poor or InSAR results are not available could readily be mapped. Thus, the issue of data scarcity in these areas can be overcome.
Indeed, the method integrating time-series InSAR and machine learning method has been shown effective in the landslide
susceptibility mapping (Ciampalini et al., 2016); however, such an integrated method has not been reported in the permafrost
stability mapping.
To illustrate the application and effectiveness of the proposed method, the permafrost stability mapping in a small area located in
the central region of the Tibetan Plateau is analyzed. The rest of this article is organized as follows. First, the study area is briefly
introduced. Second, the principle of the proposed method and the data processing are provided. Third, the ground deformation in
the study area, obtained with the time-series InSAR, is presented, and the influences of topography and vegetation coverage on
the ground deformation are studied. Fourth, the permafrost stability mapping results in the study area, obtained with the proposed
method, are presented. Fifth, the ground deformation and permafrost stability mapping results obtained are validated and
discussed. Finally, the concluding remarks are provided.

## 2. Information of the Study Area

The Tibetan Plateau has the largest permafrost area in the middle and low latitude regions of the Earth, with approximately 40%
coverage of the permafrost on the Tibetan Plateau (Zou et al., 2017; Jiang et al., 2020). According to the permafrost continuity,
the duration of frozen ground, and the maximum depth of seasonal frost penetration, the permafrost on the Tibetan Plateau can
be categorized into six types (http://westdc.westgis.ac.cn): predominantly continuous permafrost, predominantly continuous and
island permafrost, mountain permafrost, middle-thick seasonally frozen ground, thin seasonally frozen ground, and short time
frozen ground. As illustrated in Fig. 1a, the predominantly continuous permafrost is mainly located in the central and northwest
of the Tibetan Plateau, the predominately continuous and island permafrost is located on the south of the predominately
continuous permafrost, the mountain permafrost is mainly located in the north, west, and south of the Tibetan Plateau, and the
seasonally frozen ground is primarily scattered in the east of the Tibetan Plateau. Affected by the complex environmental
conditions, the responses of these six types of permafrost to climate warming can be different.

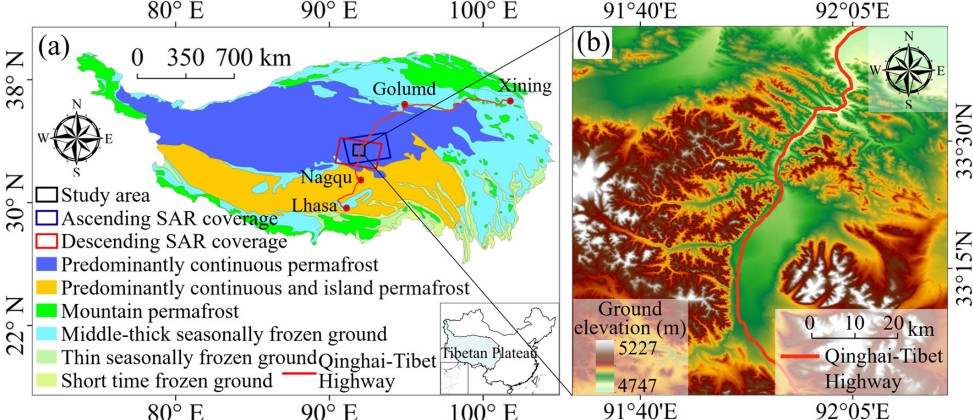

**Figure 1: General information of the study area: (a) Permafrost types on the Tibetan plateau (note: the classification results are from the Cold and Arid Regions Sciences Data Center at Lanzhou: http://westdc.westgis.ac.cn); (b) Ground elevation map of the study area**



To illustrate the application and effectiveness of the integrated method proposed, a small area located in the central region of the
Tibetan Plateau, as shown in Fig. 1a, is analyzed in this paper. The reasons for selecting this study area are summarized as
follows: 1) the topography of the study area is relatively complex and the permafrost stability mapping in this area could not be
obtained directly from InSAR analysis, as such, the random forest method may be taken as an effective and necessary
supplement to the InSAR analysis in permafrost the stability mapping; 2) although river talik exists, this area is predominately
occupied by the continuous permafrost and the permafrost stability in which has been studied by many authors (e.g., Wu et al.,
2018; Wang et al., 2019; Zhao et al., 2021), the results indicate that permafrost degradation occurs frequently in this area under
climate warming; 3) this area is covered by both ascending and descending SAR data (see Fig. 1a), as such, the permafrost
stability mapping results obtained could be validated; and 4) the Qinghai-Tibet Highway crosses this area, permafrost stability
mapping in this area will be significant for the operation of this highway.
As can be seen from Fig. 1b, the dimension of the study area is 80 km by 80 km, the topography mainly consists of the
mountainous terrain with ground elevations ranging from 4747 to 5227 m, and the Qinghai-Tibet Highway crosses the study area.
The bedrock of the study area is the red or gray sandstone and mudstone, and lacustrine deposits could also be identified in the
study area. The vegetation cover mainly consists of the alpine meadow and desert grassland. The climate is cold and dry with the
mean annual air temperature of about 4.5 °C, and the annual precipitation ranges from 300 to 400 mm. Note that the precipitation
is mainly concentrated in the rainy season (from June to August), and the heavy rainfall in the rainy season often brings about
flooding and surface erosion in the study area (Chen et al., 2012). Thus, the water content of the soil in the study area is fairly
low; and, under the effects of freeze-thaw cycles and surface runoff processes, the study area is prone to suffer from permafrost
degradation and desertification (Lu et al., 2019).

## 3. Methodology and Data Processing

In this section, the principle of the proposed method for permafrost stability mapping is presented, and the data processing
involved in this integrated method, including the time-series InSAR analysis, analysis of geometric distortion in SAR images,
and random forest method-based permafrost stability mapping, is introduced.

### 3.1 Principle of the integrated method for permafrost stability mapping

To overcome the data scarcity issue in areas where the visibility of SAR images is poor or InSAR results are not available in the
InSAR-based permafrost stability mapping, an integrated method that can take advantage of the effectiveness of InSAR analysis
(in monitoring the ground deformation in areas with good visibility of SAR images) and that of the machine learning (in
mapping the relationship between the environmental factors and the permafrost stability) is advanced in this study. The general
principle of this integrated method is illustrated in Fig. 2.
Within the context of the integrated method, the ground deformation in the concerned region is first studied with the time-series
InSAR analysis, through which an initial permafrost stability mapping is obtained. It is noted that the ground deformation, in this
initial permafrost stability mapping, cannot be available in areas with dense vegetation and water covering, due to the spatial
decoherence induced in the processing of input SAR images; whereas, the ground deformation obtained in areas with poor
visibility (of SAR images) could be problematic. Then, a screening analysis, which is based on the analysis of geometric
distortion (in input SAR images) and the coherence of InSAR analysis results, is conducted to locate the area with high-quality
ground deformation data, the screened area is termed as high-quality area while the left area is termed as low-quality area.



The studies in Ran et al. (2021) depicted that the ground deformation and the permafrost stability can be closely correlated with
the environmental factors, including the topography, land cover, land surface temperature, and distance-to-road. Note that
although the performance of the time-series InSAR technique in monitoring the ground deformation in the high-quality area and
that in the low-quality area are different, the mapping relationship between the environmental factors and the permafrost stability
in the high-quality area and that in the low-quality area could share similar feature in a specified region; indeed, this concept is
often employed in the landslide susceptibility mapping (Guzzetti et al., 2005; Gemitzi et al., 2011). The machine learning
method has been extensively adopted for mapping the relationship between the environmental factors and the landslide
occurrence. As such, the potential relationship between the environmental factors and the permafrost stability can first be trained
by the data (i.e., the ground deformation and the environmental factors) extracted in the high-quality area using the machine
learning method, and the trained relationship may be further adopted to map the permafrost stability in the low-quality area. As
an outcome, the ground deformation (and permafrost stability) in the low-quality area where the visibility of SAR images is poor
or the InSAR results are not available can be mapped from the high-quality data extracted from the initial InSAR analysis; thus,
the data scarcity issue in the InSAR-based permafrost stability mapping can be overcome, and an improved permafrost stability
mapping can be achieved.

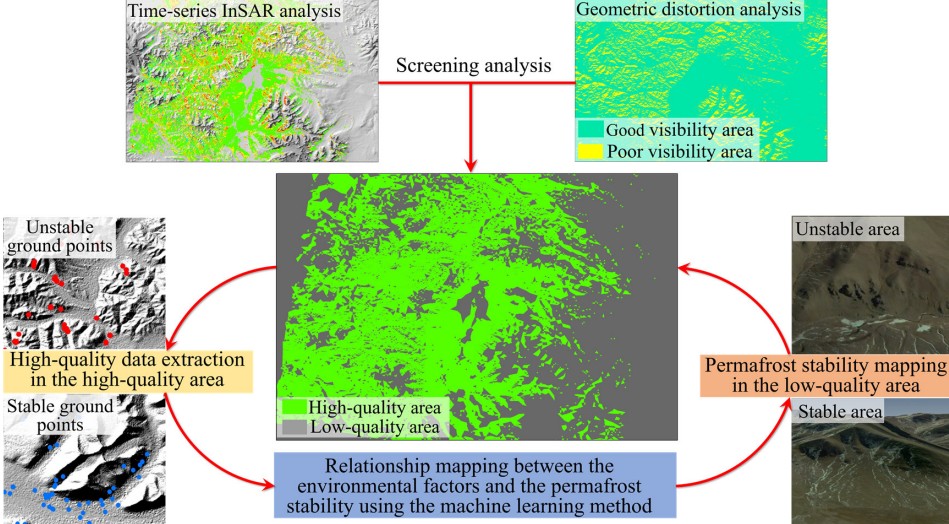


**Figure 2: Principle of the integrated method for permafrost stability mapping (© Google Earth 2019)**
**3.2 Data processing involved in the proposed method**
The time-series InSAR analysis, geometric distortion analysis of input SAR images, and random forest method-based permafrost
stability mapping mentioned above are detailed below.
**3.2.1 Time-series InSAR analysis of the ground deformation**
To analyze the ground deformations in the study area, 67 scenes of SAR images, acquired by the descending Sentinel-1 from
October 2014 to August 2020, are downloaded from the European Space Agency (https://earth.esa.int). Further, 69 scenes of
SAR images, acquired by the ascending Sentinel-1 in the same observation period are downloaded to validate the accuracy of the
permafrost stability mapping obtained from the integrated method. The boundaries of these SAR images are provided in Fig 1. In
this study, SBAS-InSAR method is employed to reduce the temporal decorrelation caused by the large time-span of the input





SAR images. The temporal baseline in time-series InSAR analyses is set at 100 days. The ground deformation in the study area
is analyzed with the following steps: 1) the signal-to-noise ratio (SNR) in the interferometric SAR images is improved with the
Goldstein radar interferogram filter (Goldstein et al., 1998); 2) the flat-earth phase and the topographic phase in the
interferometric SAR images are removed by the precise orbit determination (POD) data and the digital elevation model (DEM)
data, respectively; 3) phase unwrapping (of interferometric SAR images obtained in the previous step) is conducted with the
minimum cost flow algorithm (MCF) (Costantini and Mario 1998); 4) the residual phase component and phase ramps (of
interferometric SAR images obtained in the previous step) are removed using the ground control points (GCPs); and 5) the time-
series ground deformation along the line of sight (LOS) direction is retrieved with the inversion model (Lauknes et al., 2011). It
is noted that the DEM in the study area is obtained from the Advanced Land Observing Satellite (ALOS), which can be
downloaded from the Alaska Satellite Facility (https://asf.alaska.edu/) and the resolution is 12.5 m/pixel; and, the GCPs are
selected with the criteria that they are far away from the large defamation points and there is no phase jump.
As shown in Fig. 1b, the study area is a relatively flat and homogeneous area, and no active fault is developed. It can be rational
to assume that ground deformation in the study area is purely vertical (Liu et al., 2013). Based on the incidence angle of the
satellite LOS, the resulting LOS deformation can be easily converted to vertical ground deformation. It is noted that the accuracy
of the ground deformation obtained from InSAR analysis can be affected by the coherent pixels, and the coherence values of
which range from 0 to 1. In general, a smaller coherence value indicates that the ground deformation obtained is less reliable,
whereas, a larger coherence value indicates the ground deformation obtained is more accurate. Thus, the coherence threshold is
often adopted in InSAR analyses, and the threshold adopted ranges from 0.4 to 0.9 depending on the topographic complexity
(Wang et al., 2018a; Wang et al., 2019). In this study, the threshold value is set at 0.8 for screening the InSAR analysis results,
which is mainly determined through a preliminary sensitivity analysis; and, this value can yield accurate and sufficient ground
deformation points.
In reference to Colesanti et al. (2003) and Lu et al. (2019), the ground deformation of the permafrost on the Tibetan Plateau,
under climate warming, can be decomposed into two elements: long-term deformation (mainly caused by the increase of the
active layer thickness under climate warming) and seasonal deformation (mainly caused by the frost heave and thaw settlement
within each freeze-thaw cycle). Thus, the ground deformation of the permafrost, denoted as $S$, might be approximated as a
function of the time.
$$S(t) = a \times t + b \times \sin(\frac{2\pi}{T} \times t) + c \times \cos(\frac{2\pi}{T} \times t) + d \qquad (1)$$
where $t$ represents the time (unit: day); $T$ represents the period of a freeze-thaw cycle, which is usually set at one year (i.e., $T = 1$
year); and, $a$, $b$, $c$, and $d$ represent the model coefficients.
**3.2.2 Analysis of geometric distortion in input SAR images using the R-index model**
Note that the quality of the InSAR analysis results (i.e., ground deformation) can be greatly affected by geometric distortion in
input SAR images, which can be analyzed from the orientation parameters of the satellite LOS (i.e., incidence angle and azimuth)
and the features of the local terrain (i.e., slope and aspect). For example, the effectiveness of the InSAR analysis results can be
degraded in areas with poor terrain visibility. To locate the area with poor visibility (in SAR images) in the study area, the R-
index model (Cigna et al., 2014; Notti et al., 2014; Ren et al., 2021), which has been widely adopted for analyzing the geometric
distortions, is employed in this paper. This R-index is calculated based on the cosine of the angle between the local terrain
surface and the radar beam, as follows (Cigna et al., 2014),
$$R\text{-}index = \sin\{\theta + \arctan[\tan\alpha \times \cos(\varphi - \beta)]\} \times La \times Sh \qquad (2)$$





where $\alpha$ is the slope of the terrain; $\beta$ is the aspect of the terrain; $\theta$ is the incidence angle of the satellite LOS; $\varphi$ is the azimuth
angle of the satellite LOS; $La$ is the layover coefficient; and, $Sh$ is the shadow coefficient. The coefficients of $La$ and $Sh$ can be
calculated using the hillshade model in ArcGIS software. The geometric distortion areas in the study area can be determined with
the following criteria: 1) if R-index is greater than or equal to $\sin(\theta)$ (i.e., R-index $\geq \sin(\theta)$), the related area is categorized as an
area with good visibility, and no geometric distortion exists; 2) if R-index is between 0 and $\sin(\theta)$ (i.e., $0 < $ R-index $ < \sin(\theta)$), the
related area is categorized as a foreshortening region, and geometric distortion exists; and 3) if R-index is not positive (i.e., R-
index $\leq 0$), the related area is categorized as a layover or shadow region, and geometric distortion exists. In this study, the area
with geometric distortions (i.e., foreshortening, layover, and shadow regions) is considered as an area with poor visibility. From
there, the high-quality area, which is defined as the intersection of the area with InSAR deformation points and the good
visibility area (no geometric distortion), in the study area can be located; whereas, the left area is categorized as the low-quality
area.
**3.2.3 Random forest method-based permafrost stability mapping**
As discussed above, the relationship mapping between the environmental factors and the permafrost stability is fairly similar to
that between the environmental factors and the landslide occurrence. It is noted that there are various approaches for mapping the
relationship between the environmental factors and the landslide occurrence, such as the artificial neural network (Lee et al. 2004;
Gong et al., 2021), decision tree (Pradhan 2013), frequency ratio (Ozdemir and Altural 2013), and fuzzy assessment (Gemitzi et
al., 2011; Gong et al., 2021). These methods can readily be adopted for mapping the relationship between the environmental
factors and the permafrost stability. In this study, random forest method (Bureau et al., 2003) is adopted for the relationship
mapping between the environmental factors and the permafrost stability. In the context of the random forest method, the
technique of bootstrap resampling is used for extracting bootstrap samples from the original samples, each bootstrap sample is
then modeled by a decision tree, and the predictions obtained from the multiple decision trees are finally combined. As such, the
issues caused by the outliers in the prediction, overfitting, and data missing in the training samples can be overcome. Indeed, the
method of random forest has been extensively adopted for the landslide susceptibility mapping (Ciampalini et al., 2016). Note
that the selection of the number of decision trees plays a vital role in the prediction accuracy of the trained random forest model.
For example, an insufficient number of decision trees may lead to the reduced accuracy of the model prediction, whereas, an
excessive number of decision trees may cause data redundancy. Based on a tradeoff analysis between prediction accuracy and
data redundancy, the number of decision trees in this study is set up as 400.
The analyses in Michaelides et al. (2019) and Chen et al. (2020) indicated that the ground deformation (and permafrost stability)
can be greatly affected by the vegetation coverage (i.e., NDVI) and the topography factors of ground elevation and slope
orientation. The studies in Deluigi et al. (2017) depicted that the ground deformation could also be influenced by the other
topographic factors, such as slope and curvature; and, the analysis in Qin et al. (2020) depicted that the land surface temperature
is a good indicator for analyzing the permafrost stability. Further, the land cover plays a vital role in influencing the permafrost
stability (Beck et al., 2015). Apart from the factors discussed above, the permafrost stability might also be degraded by the
engineering activities. For example, the construction and operation of the Qinghai-Tibet Highway have led to an obvious
degradation of the permafrost along this highway (Yu et al., 2013).
Under such circumstances, eight environmental factors, including the ground elevation, aspect, slope, curvature, land cover,
NDVI, land surface temperature, and distance to the Qinghai-Tibet Highway, are extracted in the study area for mapping the
permafrost stability. Here, the topography factors (i.e., ground elevation, aspect, slope, and curvature) are calculated from the
ALOS DEM (https://asf.alaska.edu/), the land cover is generated from GlobeLand30 product (http://www.globallandcover.com/),



the NDVI and land surface temperature are obtained from the analysis of Landsat-8 synthesized cloudless images conducted on
the Google Earth Engine cloud platform (http://earthengine.google.org/), and the distance to the Qinghai-Tibet Highway is
generated by the Euclidean distance function in ArcGIS software. Plotted in Fig. 3 are the environmental factors extracted in the
study area, which are resampled into the 100 m by 100 m spatial grids.
A multicollinearity analysis indicates that the environmental factors shown in Fig. 3 are independent of each other. For ease of
screening stable and unstable ground points in the initial InSAR analysis results, threshold values of the ground deformation rate
are prespecified; and, the determination of these threshold values is detailed in the section of results. In order to train the
relationship between the environmental factors and the permafrost stability, 80% of the high-quality ground points, which are
extracted in the high-quality area and screened according to the threshold values of the ground deformation rate, are taken as the
training samples while the left 20% of the high-quality ground points are taken as the testing samples.

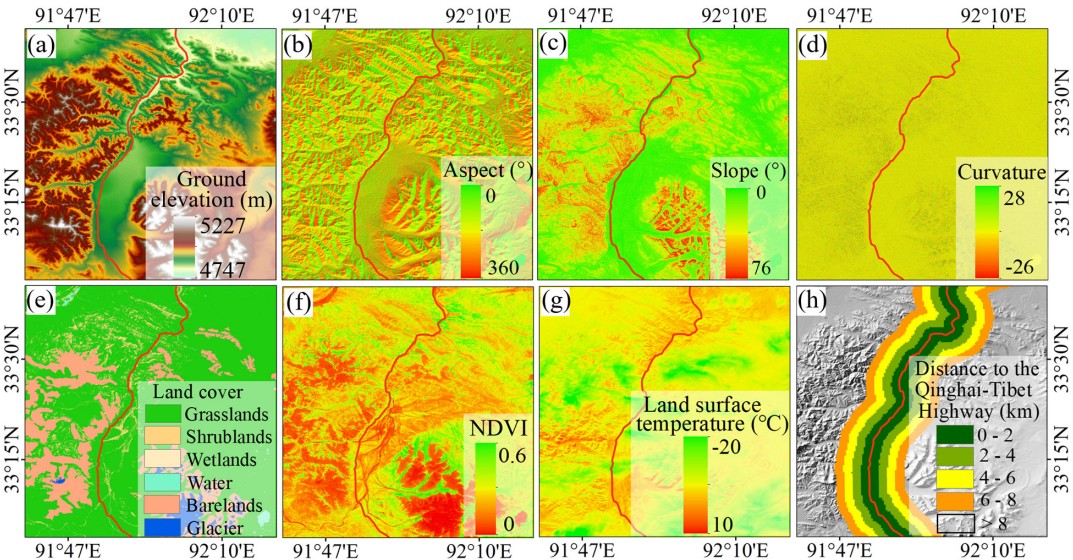


**Figure 3: Environmental factors extracted in the study area: (a) Ground elevation; (b) Aspect; (c) Slope; (d) Curvature;
(e) Land cover; (f) NDVI; (g) Land surface temperature; (h) Distance to the Qinghai-Tibet Highway**

## 4. Results

The ground deformations and the permafrost stability mapping results in the study area, obtained with the time-series InSAR
analysis and the proposed method, respectively, are presented in this section. Meanwhile, the influences of topography and
vegetation coverage on the ground deformation are investigated.

### 4.1 Results of the ground deformation with time-series InSAR analysis

The ground deformations obtained in the study area are presented below, and the influences of topography and vegetation
coverage are discussed.

#### 4.1.1 Ground deformations obtained in the study area

Fig. 4a shows the vertical ground deformation rate in the study area obtained from October 2014 to August 2020. As can be seen,
the ground deformation rate ranges from -58 mm/year to 29 mm/year; and, the regions with permafrost instability, indicated by
the area with large deformation rates (either subsidence or upheaval), are mainly distributed in the valley areas with low altitude.





In addition, the ground deformation mainly takes place in the west-facing slopes (see the comparison in Fig. 4b-4c), partially
because the input SAR images are collected by the descending satellite.

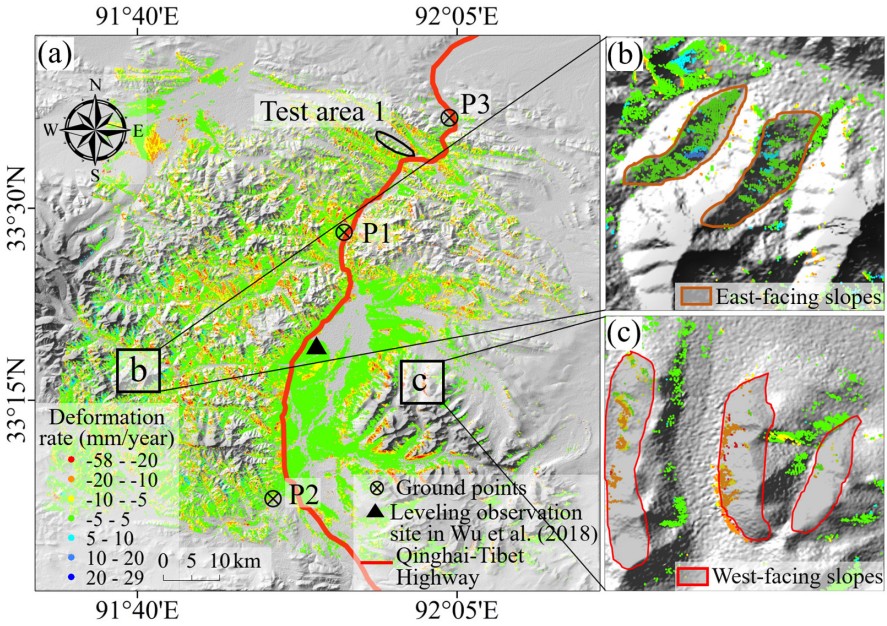

**Figure 4: InSAR analysis results of the ground deformation in the study area: (a) Vertical ground deformation rate from October 2014 to August 2020; (b) Ground deformation in east-facing slopes; (c) Ground deformation in west-facing slopes**

As described above, the ground deformation of the permafrost area can be decomposed into two elements: long-term
deformation and seasonal deformation. Here, the ground deformations at three points (in terms of points P1, P2, and P3 in Fig. 4a)
are used to analyze the ground deformations using the empirical model established in Eq. (1). According to the ground
deformations monitored from October 2014 to August 2020, the model coefficients can be estimated with the least square
method. The resulting models, as shown in Fig. 5a, yield R-squares of 0.50, 0.80, and 0.83, indicating the ground deformations in
permafrost areas can be well captured by the empirical model shown in Eq. (1). The plots in Fig. 5a also indicate that the
permafrost exhibit upheaval in the frozen season (from September to March, attributed to the frost heave of active layer) and
exhibit subsidence in the thawing season (from April to August, attributed to the thaws of active layer). Hence, the maximum
ground settlement of the permafrost occurs around August each year. Fig. 5b depicts that the seasonal deformations (calculated
as the total deformation minus the long-term deformation, expressed as $S(t) - a \times t$) tend to be negatively correlated with the air
temperature, which confirms that the seasonal deformation is mainly caused by the frost heave and thaw settlement within each
freeze-thaw cycle, such similar phenomena were also observed in Zhao et al. (2016) and Lu et al. (2019).



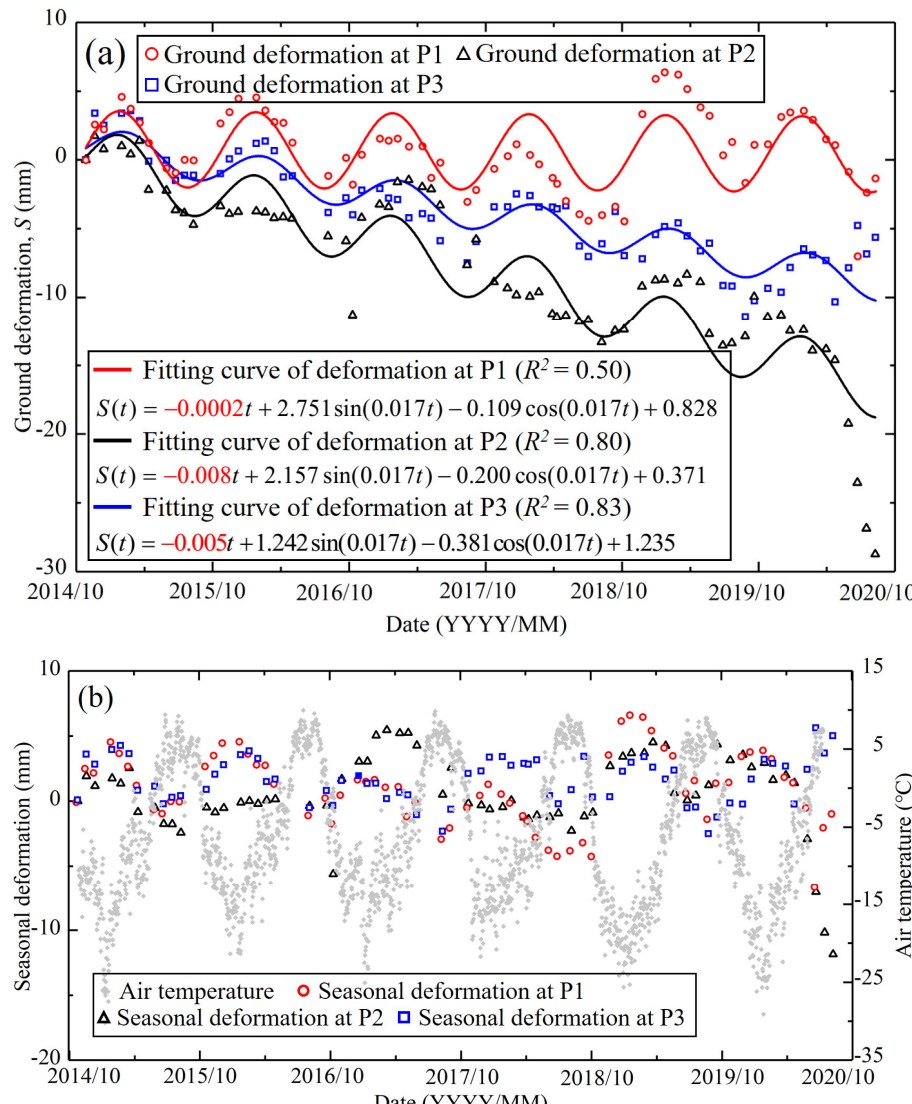



**Figure 5: Fitting analysis of the ground deformation in the period from 2014 to 2020: (a) Fitting analysis between the ground deformation and the time; (b) Correlation analysis between the seasonal deformation and the air temperature (note: the air temperature is from the 2.0 m air temperature dataset of the European Centre for Medium-Range Weather Forecasting Fifth-generation Reanalysis (ECMWF ERA5))**

Fig. 6 depicts the maximum ground deformations of the study area occurred during the thawing periods in 2015, 2017, 2018, and
2019. It is found that both magnitude and scope of the maximum ground deformation continue to increase from 2015 to 2019,
which is quite evident in the northern region of the study area, though there may be some exceptions due to the accuracy of the
ground deformation results. Note that the ground elevation of this study area tends to decrease from south to north, except that
along the Qinghai-Tibet Highway, as shown in Fig. 1b. Therefore, the region with a lower altitude has a higher risk of permafrost
instability or degradation, this inference is in general agreement with the observations in Lu et al. (2019) and Huang et al. (2020).
For example, more thermokarst lakes, retrogressive thaw slumps, and failed slopes are detected in the regions with low altitudes
(Lu et al., 2019; Huang et al., 2020).




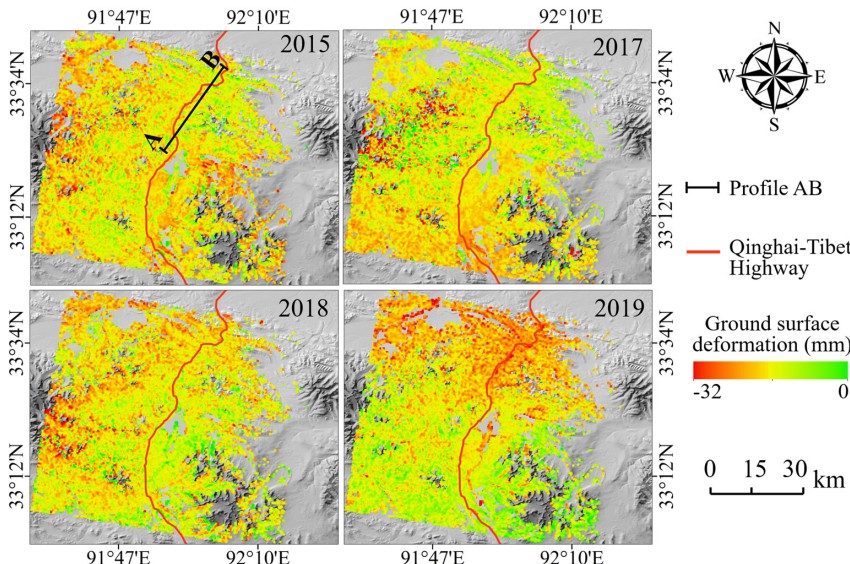


**Figure 6: The maximum ground deformations of the study area occurred during the thawing periods in 2015, 2017, 2018, and 2019**

**4.1.2 Influence of the topography on the ground deformation**

It is worthwhile mentioning that the transfers of water and heat in the frozen soil could be strongly affected by environmental factors, and the transfers of water and heat can lead to phase changes of the water in the active layer, which consequently results in ground deformation and affects the permafrost stability (Zhao et al., 2016). In other words, the ground s deformation in permafrost areas is highly related to the ice or water content in the active layer (Arenson et al., 2016); however, the ice or water content of the soil in a large area is challenging to monitor. Thus, only the influences of topography and vegetation coverage, which have great influences on the distribution of the ice or water content in the soil (Michaelides et al., 2019; Chen et al., 2020), on the ground deformation are studied here to analyze the influences of the environmental factors on the permafrost stability.

It is known that the watershed and river network in an area are mainly determined by the topography (Chen et al., 2020), thus, the distribution of the water content in the ground is strongly affected by the local topography. To analyze the influence of the topography on the ground deformation of the permafrost, the river network in the study area is generated from the DEM, which is then superimposed on the average thaw subsidence took place in the period from 2015 to 2019, as illustrated in Fig. 7a. As can be seen, the regions with large thaw subsidence are mainly located in the river valleys where the soil is typically fully saturated and the ground ice is rich (Zhang and Wu 2012; Chen et al., 2020); whereas, the regions with small thaw subsidence are mainly located at the hill ridges where the water content in the ground is relatively low. Next, the influence of the topography on the ground deformation is investigated based on the data collected along the profile AB (see Fig. 7a) and the study results are depicted in Fig. 7b-7c. Fig. 7b depicts the relationship between the acquired average thaw subsidence and the ground elevation. Fig. 7c shows that the change in the magnitude of the thaw subsidence is in general agreement with that in the ground elevation, and a larger ground elevation tends to yield smaller thaw subsidence but there are exceptions. For example, the thaw subsidence in Zone I matches the ground elevation well while that in Zone II cannot match the ground elevation. A detailed survey of the topography suggests that Zone I is located in a river valley while Zone II is located on a north-facing slope (see Fig. 7d). It must be noted that although the ground elevation in Zone II is lower than that in Zone I, the solar radiation in Zone II can be much





324  weaker, and thus the ice in the ground is more difficult to melt. From there, the ground elevation and slope orientation play a

325  vital role in determining the influence of the topography on the ground deformation (and permafrost stability).

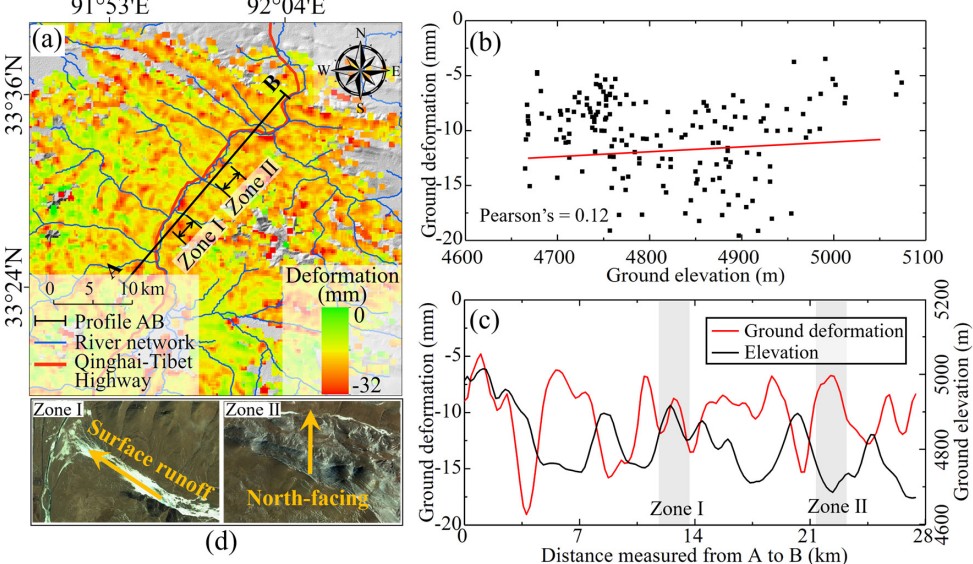

326

**Figure 7: The influence of the topography on the ground deformation of the permafrost: (a) Average thaw subsidence of the study area took place in the period from 2015 to 2019; (b) Relationship between the thaw subsidence and the ground elevation along profile AB; (c) Variations of the thaw subsidence and the ground elevation with the distance measured from A to B along profile AB; (d) A detailed survey of the topography (Zone I is located in a river valley while Zone II is located on a north-facing slope, © Google Earth 2019)**

### 4.1.3 Influence of the vegetation coverage on the ground deformation

The vegetation coverage is also taken as an important index of the water content in the ground (Michaelides et al., 2019). The
influence of the vegetation coverage on the ground deformation in the study area is herein investigated. In reference to Fig. 8a,
the vegetation coverage in the study area is dominated by grasslands and bare lands. The NDVI is employed in this study to
represent the vegetation coverage, and a larger NDVI value signals denser vegetation coverage. The influence of the vegetation
coverage (i.e., NDVI) on the ground deformation (i.e., the average thaw subsidence took place in the period from 2015 to 2019,
see Fig. 7a) is then studied based on the data collected along the profile AB, and the results are illustrated in Fig. 8b-8c. Similar
to that in Fig. 7c, the change in the magnitude of the thaw subsidence is in good agreement with that in NDVI, and a larger
NDVI value tends to result in smaller thaw subsidence, partially due to the protective effect of the vegetation coverage (on the
ground) against the ice melting (in the ground). The studies in Michaelides et al. (2019) confirmed that the microbial activity in
the ground would increase and the permafrost degradation would aggravate when losing the protection of the vegetation.
The plots in Fig. 8(c) also indicate that for the same NDVI value, the thaw subsidence might vary with the position along the
profile AB, implying that the influence of the vegetation coverage on the ground deformation can be rather complicated. In most
cases, the relationship between permafrost stability and vegetation may be interdependent or symbiotic (Anderson et al., 2019;
Jin et al., 2021). On one hand, the vegetation coverage could shade direct sunshine in summer and meanwhile intercept the
snowfall in winter, as such, the vegetation helps cool the ground and thus protect the underlying permafrost. On the other hand,
the shallow thickness of the active layer and the low temperature of the ground can prevent the growth of the vegetation.



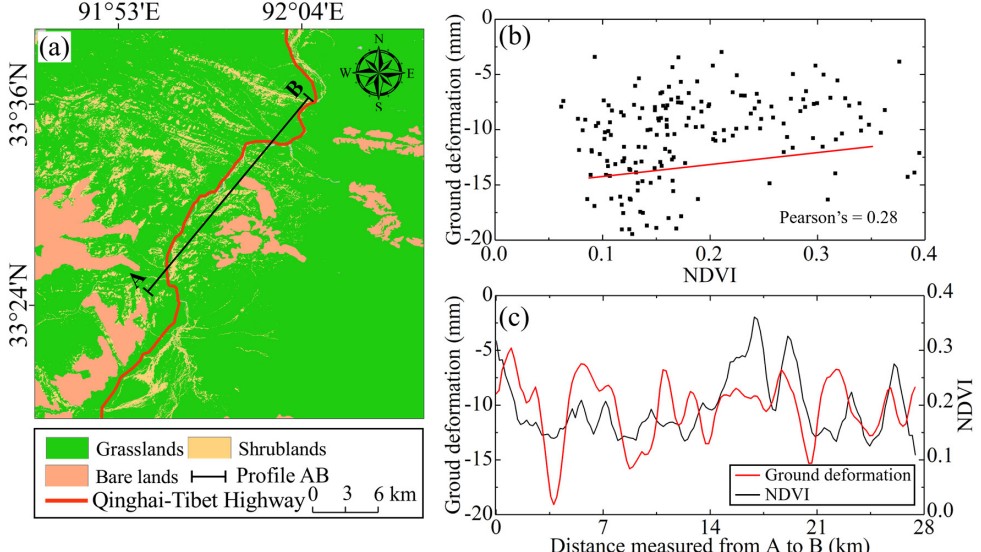


**Figure 8: The influence of the vegetation coverage on the ground deformation of the permafrost: (a) Vegetation coverage in the study area; (b) Relationship between the thaw subsidence and the NDVI along profile AB; (c) Variations of the thaw subsidence and the NDVI with the distance measured from A to B along profile AB**

**4.2 Results of permafrost stability mapping with the proposed method**

The screening results of the high-quality and low-quality areas, based on the time-series InSAR analysis and geometric distortion analysis, and the results of permafrost stability mapping, derived with the proposed method, are detailed below.

**4.2.1 Screening results of the high-quality and low-quality areas**

As mentioned previously, the coherence threshold is set at 0.8 in the processing of SAR images. As a result, only the high-quality InSAR deformation points could be kept in the initial InSAR analysis of the ground deformations; whereas, the InSAR deformation points the coherence of which is less than the threshold value of 0.8 are not displayed. Thus, the initial map of the obtained ground deformation cannot cover the entire study area, as shown in Fig. 4a. Here, the area that does not have ground deformation is categorized as the low-quality area.

The geometric distortion analysis results are sketched in Fig. 9a. As can be seen, most area can be categorized as the good visibility area while the east-facing slopes are mainly located in the regions with geometric distortions (see Fig. 9b). The area with geometric distortions is then categorized as the low-quality area. From there, the high-quality area, which is defined as the intersection of the area with InSAR deformation points (see Fig. 4a) and the good visibility area (see Fig. 9a), in the study area can be located. Fig. 9c depicts the zonation of the high-quality and low-quality areas in the study area. Here, the ground deformation monitored in the high-quality area is reliable while that monitored in the low-quality area can be problematic.

To screen the stable and unstable ground points in the initial InSAR analysis results, the threshold values of the ground deformation rate in this study are set at ±0.15 mm/year and -40 mm/year. That is to say, the ground point with a ground deformation rate smaller than -40 mm/year and obvious unstable characteristics is classified as an unstable ground point, while that with a deformation rate ranges from -0.15 mm/year to 0.15 mm/year and no obvious unstable characteristics is classified as a stable ground point. In the high-quality area located, a total number of 586 unstable ground points are recognized, as shown in Fig. 9c. To avoid the potential bias in the selection of samples, an equal number of stable ground points are identified in the high-



quality area. The distribution of the selected stable ground points is also displayed in Fig. 9c. In summary, a total number of
1,172 high-quality ground points are located in the initial InSAR analysis results. As the ground-based deformation measurement
is fairly limited in the study area, the high-quality ground points obtained from InSAR are mainly validated through visual
interpretations of the Google Earth images; whereas, verifications of the time-series InSAR analysis results are presented in the
next section.

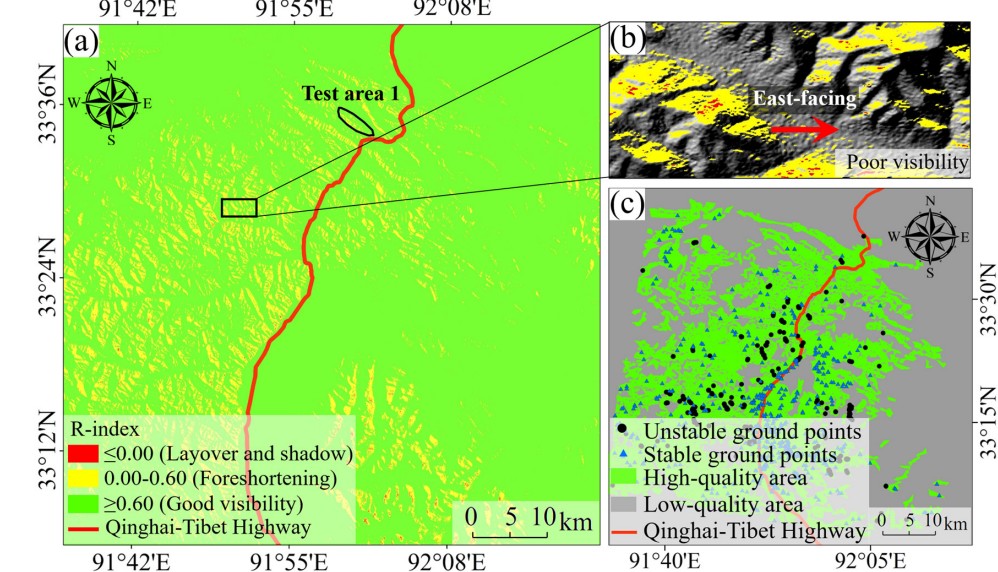

**Figure 9: Screening analysis of the initial InSAR analysis results of ground deformation in the study area: (a) Geometric**
**distortion analysis results; (b) A detailed survey of the geometric distortions on an east-facing slope; (c) Zonation of the**
    **high-quality and low-quality areas**

**4.2.2 Permafrost stability mapping in the low-quality area with random forest method**
In order to train the relationship between the environmental factors and the permafrost stability, 80% of the 1,172 high-quality
ground points screened above (i.e., 80% of 586 unstable ground points and 80% of 586 stable ground points) are taken as the
training samples while the left 20% of the 1,172 high-quality ground points are taken as the testing samples. The outcome of the
permafrost stability mapping by the trained random forest model is a value ranging from 0 to 1, which indicates the probability
of permafrost stability. For example, 0 represents permafrost instability while 1 represents permafrost stability. For ease of visual
interpretation, this value is then categorized into five classes of permafrost stability (i.e., very low, low, medium, high, and very
high) with the Jenks optimization method (Jenks 1967).
Fig. 10a depicts the results of the permafrost stability mapping in the study area with the trained random forest model. The area
with very low and low permafrost stability is mainly distributed along the Qinghai-Tibet Highway. The receiver operating
characteristics (ROC) curve, which is widely adopted for evaluating the overall prediction accuracy of landslide susceptibility
models and the associated zonation maps (Mason and Graham 2002), is employed in this study for evaluating the mapping
accuracy of the trained random forest model. The ROC curve plots the true positive rate on Y-axis and the false positive rate on
X-axis. The area under the curve (AUC) measures the probability of correct classification, and an AUC value close to 1 indicates
high mapping accuracy. The calculated AUC value of the permafrost stability mapping results in the study area is 97.5%,
indicating high mapping accuracy of the trained random forest model. Meanwhile, among the 234 testing samples, 82.05% of the




unstable ground points are located in areas with very low and low permafrost stability. Therefore, the mapping accuracy of the
trained random forest model can be validated.
The relative importance of the environmental factors to the permafrost stability could be studied as a byproduct of the training of
the random forest model, and the analysis results are plotted in Fig. 11. The relative importance of each environmental factor to
the permafrost stability is evaluated here by the indexes of mean decrease accuracy (MDA) and mean decrease Gini (MDG),
which can be calculated according to the reduction of the prediction accuracy when values of this environmental factor in a
decision tree are permuted randomly (Calle and Urrea 2010). In general, larger values of these two indexes (i.e., MDA and MDG)
could signal greater importance of the related environmental factor. As can be seen from Fig. 11, the permafrost stability is most
affected by the slope and the aspect while is least affected by the curvature; and, the left environmental factors yield similar
importance in the permafrost stability mapping.
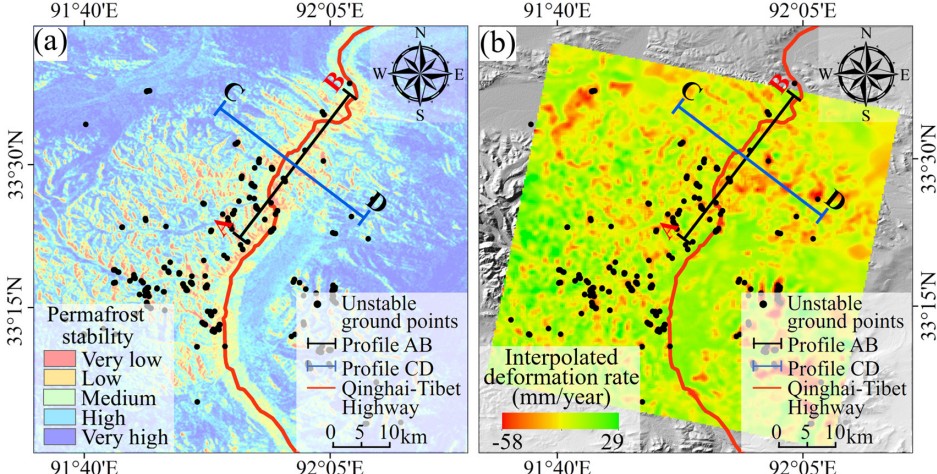

**Figure 10: The permafrost stability mapping in the study area: (a) Results of the permafrost stability mapping with the**
**trained random forest model; (b) Ground deformation rate obtained by the Kriging interpolation of initial InSAR**
**analysis**
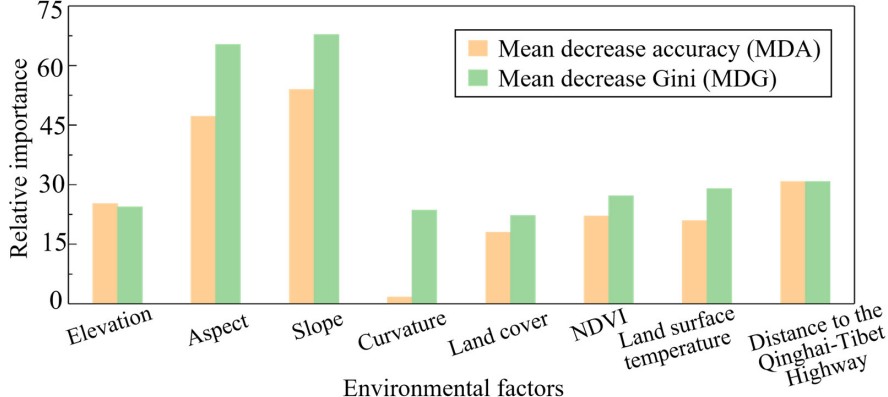

**Figure 11: The relative importance of the environmental factors to the permafrost stability**
**5. Verifications and Discussions**

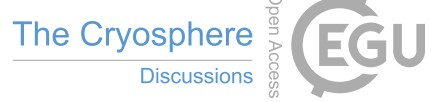

The ground deformations and the permafrost stability mapping results, obtained with the time-series InSAR analysis and the
proposed method, respectively, are validated in this section. Further, the superiority of the proposed method for permafrost
stability mapping over the sole application of InSAR analysis is discussed.

**5.1 Verifications of the ground deformations obtained with InSAR analysis**

As formulated above, the high-quality ground points (i.e., the basic inputs to the proposed permafrost stability mapping) are
derived from the time-series InSAR analysis. Thus, verification of the accuracy of the time-series InSAR analysis results is vital.
Note that the InSAR analysis results and the field measurements often cover different temporal and spatial scales, a direct
verification by the field measurements might be impossible (Chen et al., 2020). Further, the ground-based deformation
measurement is fairly limited in the study area. In this study, the time-series InSAR analysis results are mainly verified through
comparing with the leveling data and InSAR analysis results outlined in Wu et al. (2018). The location of the leveling
observation site is labeled in Fig. 4a. Fig. 12 shows the InSAR analysis results obtained in this study, together with the leveling
data and InSAR analysis results obtained in Wu et al. (2018). As can be seen, the InSAR analysis results obtained in this study
are in general agreement with the leveling data and InSAR analysis results obtained in Wu et al. (2018), even though an
inconsistency exists in the frozen season from 2015 to 2016. In our study, frost heave is observed in this frozen season; whereas,
the thaw settlement was detected in Wu et al. (2018). The InSAR analysis results obtained in this study appear to be more
consistent with the available knowledge of ground deformations in the study area, than those outlined in Wu et al. (2018).
Next, the relationships between the ground deformations and the air temperature are analyzed to further verify the effectiveness
of the time-series InSAR analysis results obtained in this study. In general, there are negative correlations between the ground
deformations and the air temperature (Zhao et al., 2016). Here, three ground points (in terms of points P1, P2, and P3 in Fig. 4a)
are studied, and the resulting relationships between the ground deformations and the air temperature are plotted in Fig. 13. As
expected, the ground deformations in the thawing seasons are large (caused by thaw settlement) while those in the frozen seasons
are small (caused by frost heave); and, the air temperature in the thawing seasons is high while that in frozen season is low. From
there, the accuracy of the time-series InSAR analysis results could be qualitatively validated.

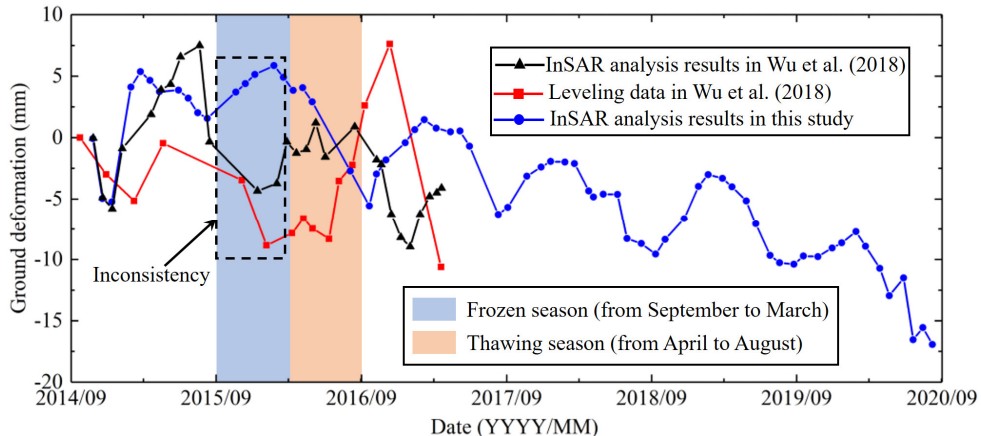


**Figure 12: Verifications of the time-series InSAR analysis results with the leveling data and InSAR analysis results**
**obtained in Wu et al. (2018)**



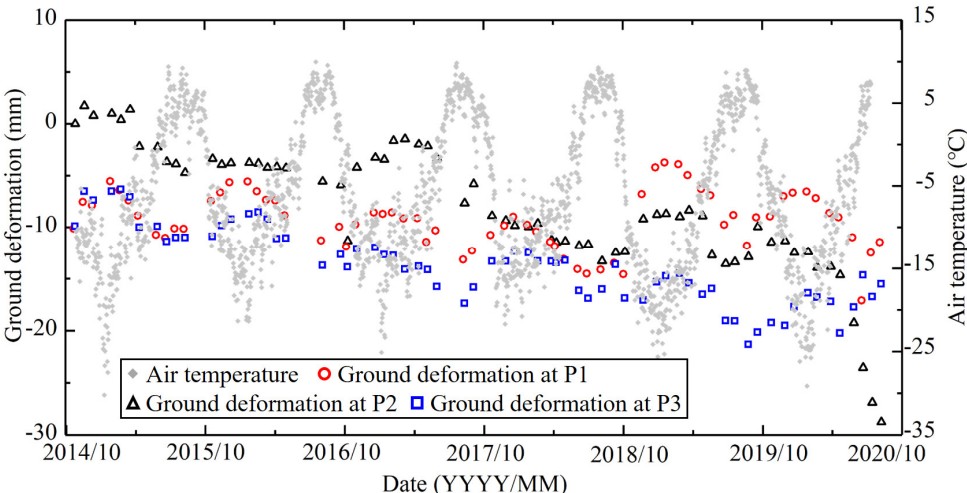

**Figure 13: Relationships between the ground deformations and the air temperature at points P1, P2, and P3**

**5.2 Verifications of permafrost stability mapping with ascending Sentinel-1 SAR images**

To validate the effectiveness of the permafrost stability mapping results obtained with the proposed method, the ground deformations in the study area are further analyzed with the ascending Sentinel-1 SAR images, and the results are illustrated in Fig. 14. Fig. 14a shows the ground deformation rate obtained from the ascending Sentinel-1 SAR images, and Fig. 14b shows the geometric distortion analysis results of the ascending SAR images. As can be seen, the ground deformation rates in the study area obtained from the ascending SAR images are in general agreement with those obtained from the descending SAR images (see Fig. 4a); and, in some areas (e.g., Test area 1 shown in Fig. 4a and 14a), the ground deformations could be obtained from ascending SAR images while cannot be obtained from the descending SAR images. The plot in Fig. 14b suggests that most regions in the study area can be categorized as the good visibility area. Hence, the ground deformations obtained from the ascending SAR images may be adopted for verifying the permafrost stability mapping results in Fig. 10a, which are derived from the descending SAR images using the proposed method.

To verify the permafrost stability mapping results shown in Fig. 10a, the permafrost stability mapping results obtained with the proposed method and the ground deformations obtained from the ascending SAR images (see Fig. 14a) are compared in Test area 1, and the comparison results are illustrated in Fig. 15. The permafrost stability in the bottom-right corner of this test area is low and very low (see Fig. 15a). Note that this corner is mainly occupied by the good visibility area in the ascending SAR images (see Fig. 14b) while occupied by the geometric distortion area in the descending SAR images (see Fig. 9a); thus, the ground deformations in this corner obtained from the ascending SAR images are reliable while those obtained from the descending SAR images can be problematic. Fig. 15b shows the ground deformations in this corner obtained from the ascending SAR images, and Fig. 15c shows those obtained from the descending SAR images. In Fig. 15b, the bottom-right corner of Test area 1 shows a trend of subsidence, which is in general agreement with the low or very low permafrost stability depicted in Fig. 15a; whereas, only limited points with ground deformations can be obtained from the descending SAR images (see Fig. 15c). From there, the accuracy of the permafrost stability mapping results obtained with the proposed method could be qualitatively validated. Further, the comparisons in Fig. 15 may also indicate that the integration of descending and ascending SAR images can provide an alternative for improving the permafrost stability mapping in some complex areas.



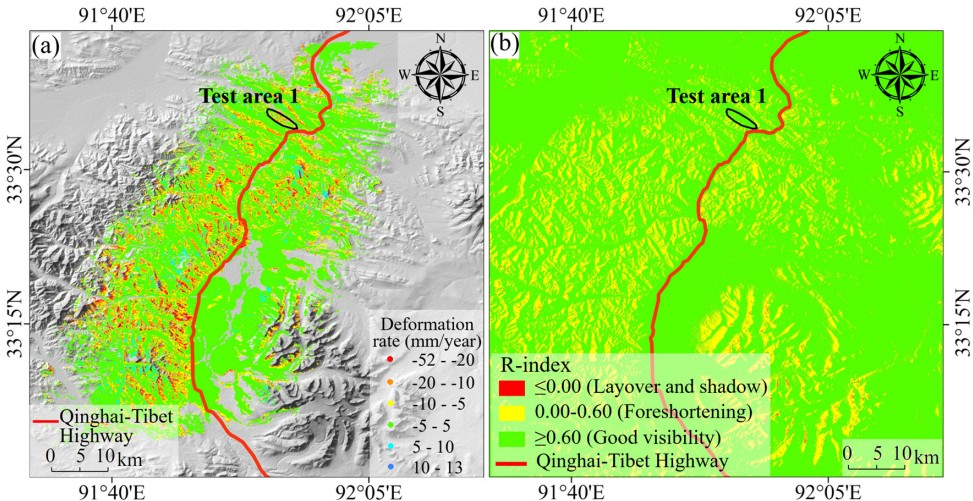

**Figure 14: InSAR analysis results in the study area obtained from the ascending Sentinel-1 SAR images: (a) Vertical ground deformation rate from October 2014 to August 2020; (b) Geometric distortion analysis results**

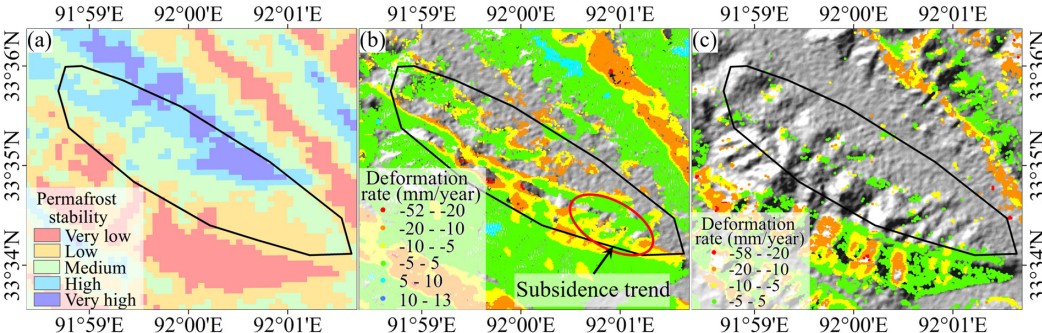

**Figure 15: Comparisons between the permafrost stability mapping results in Test area 1 obtained with the proposed method and the ground deformations obtained from the ascending SAR images: (a) Permafrost stability mapping results obtained with the proposed method; (b) Vertical ground deformation rates obtained from the ascending SAR images; (c) Vertical ground deformation rates obtained from the descending SAR images**

**5.3 Superiority of the proposed method over the sole application of InSAR analysis**

To demonstrate the superiority of the proposed method over the sole adoption of InSAR analysis in permafrost stability mapping, comparative analyses between the permafrost stability zonation obtained by the proposed method and the ground deformation rate obtained from the InSAR analysis (with descending Sentinel-1 SAR images as inputs) are conducted. For ease of comparisons, the ground deformation rate in the area where the InSAR analysis results are not available is interpolated here using the Kriging method (Matheron 1976), and the results of this interpolation are plotted in Fig. 10b. As can be seen, the permafrost stability zonation obtained by the proposed method is in general agreement with the ground deformation rate obtained by InSAR analysis (see Fig. 10a-10b), however, due to the interpolated accuracy of the ground deformations, there are exceptions. For example, the permafrost stability zonation is not consistent with the ground deformation rates in Zones III, IV, V, and VI. Fig. 16 shows a detailed comparison between the permafrost stability zonation obtained by the proposed method and the ground deformation rate obtained by InSAR analysis along the profiles AB and CD (note: these two profiles are depicted in Fig. 10a-10b).



It can be seen from Fig.16a-16b that Zones III and VI are located in the area with medium and high permafrost stability,
according to the permafrost stability mapping obtained by the proposed method. The permafrost stability in Zones III and VI can
be visually confirmed by the Google Earth images, as depicted in Figure 16c. However, the ground deformation rate (obtained by
a combination of InSAR analysis and Kriging interpolation) in Zones III and VI could reach -10 mm/year, indicating instability
of the permafrost. Similarly, according to the permafrost stability zonation obtained by the proposed method, Zones IV and V are
located in the area with low permafrost stability. In reference to the Google Earth images shown in Figure 16c, the stability of the
permafrost in Zones IV and V is fairly poor, evidenced by retrogressive thaw slumps and failed slopes. However, the ground
deformation rate (obtained by the combination of InSAR analysis and Kriging interpolation) in Zones IV and V is larger than -5
mm/year, indicating stability of the permafrost. Hence, the proposed method is shown more effective in the permafrost stability
mapping, compared to the sole adoption of InSAR analysis, and the data scarcity issue of InSAR analysis in the low-quality area
could be overcome.

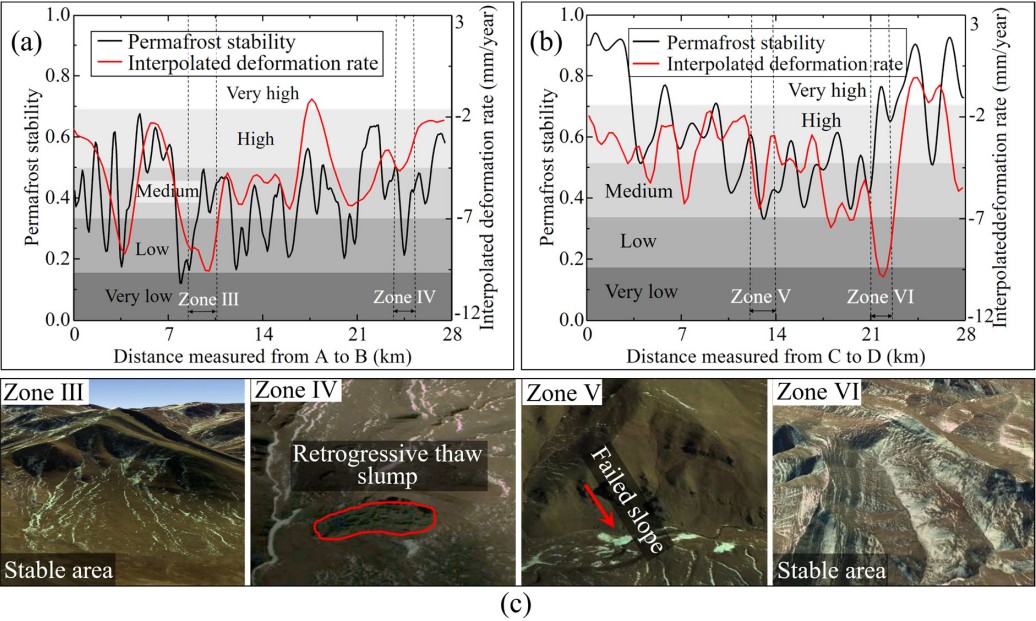

**Figure 16: Comparisons between the proposed method and the sole application of InSAR analysis: (a) Permafrost**
**stability zonation obtained by the proposed method versus the ground deformation rate obtained by the Kriging**
**interpolation of initial InSAR analysis along profile AB; (b) Permafrost stability zonation obtained by the integrated**
**method versus the ground deformation rate obtained by the Kriging interpolation of initial InSAR analysis along profile**
**CD; (c) A detailed survey of the permafrost stability in Zones III, IV, V, and VI with the Google Earth images (note:**
**Zones III and VI are located in the area with medium and high permafrost stability, Zones IV and V are located in the**
**area with low permafrost stability, © Google Earth 2019)**
**6. Concluding Remarks**
This paper presents a method that integrates InSAR and random forest method for an improved permafrost stability mapping on
the Tibetan Plateau, with which the data scarcity of InSAR analysis in the low-quality area (i.e., where InSAR analysis results
are not available due to the coherence of InSAR analysis results and geometric distortions in input SAR images) could be
overcome. To demonstrate the application of this proposed method, the permafrost stability mapping in a small area located in
the central region of the Tibetan Plateau is studied, the results obtained are validated through qualitative verifications, and



comparative analyses are conducted to illustrate the superiority of this integrated method over the sole adoption of InSAR
analysis in the permafrost stability mapping. Based upon the results presented, the following conclusions are reached.
1) The initial InSAR analysis of the ground deformation (in the study area) shows that the maximum ground settlement of the
permafrost occurs around August each year, due to the frost heave of the active layer in the frozen season and subsidence in the
thawing season; and, both magnitude and scope of the ground deformation tend to increases from 2015 to 2019, which might be
taken as a sign of the deterioration of the permafrost. The initial InSAR analysis also confirms that the ground deformation is
strongly affected by the topography and vegetation coverage.
2) According to the analysis of geometric distortion and coherence of InSAR results, the high-quality area could be recognized,
in which high-quality ground points can readily be located based on the threshold values of the ground deformation rate. The
permafrost stability and associated environmental factors of these high-quality ground points can then be extracted for the
permafrost stability mapping in the low-quality area. The random forest-based mapping analysis suggests that the permafrost
stability (in the study area) is most affected by the slope and aspect while is least affected by the curvature; and, the factors of the
ground elevation, land cover, NDVI, land surface temperature, and distance to the highway yield similar importance in the
permafrost stability mapping.
3) The validation analysis of the obtained permafrost stability zonation, which is based on the ROC curve and the unstable
ground points in the testing samples, indicates that this integrated method could yield high mapping accuracy in the study area.
Through qualitative verifications, the ground deformations and the permafrost stability mapping results, obtained with the time-
series InSAR analysis and the proposed method, respectively, could be validated. Compared to the sole adoption of InSAR
analysis, this integrated method is shown more effective in the permafrost stability mapping in the study area; meanwhile, the
issue of data scarcity of InSAR analysis in the low-quality area could be overcome.
It should be mentioned that although the proposed method has been shown promising in the permafrost stability mapping in the
study area, there is room for improvement. For example, research to further validate the permafrost stability zonation with
ground-based measurements is warranted; and, InSAR analysis in the study is based on the Sentinel-1 C-band SAR images with
a 6-day revisiting time, the effectiveness of which is often degraded in the mountainous and vegetated areas, research on data
fusion methods that could integrate different sources of SAR images is also warranted.
**Data availability**
All raw data can be provided by the corresponding authors upon request.
**Author contribution**
FZ and WG planned the campaign; FZ, WG, TR, and JC collect and prepare the data; FZ, WG, TR, and TL analyze the data;
FZ and WG wrote the manuscript draft; WG and HT edited the manuscript and provide the financial support.
**Competing interests**
The authors declare that they have no conflict of interest.
**Acknowledgments**





The financial support provided by the National Natural Science Foundation of China (Grants No. 41977242 and No. 41702294), the Major Program of National Natural Science Foundation of China (No. 42090055), and the Fundamental Research Funds for the Central Universities, China University of Geosciences (Wuhan) (Grant No. CUGGC09) is acknowledged.

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
