# Peer review of "Permafrost Stability Mapping on the Tibetan Plateau by Integrating Time-series InSAR and Random Forest Method"

_The Cryosphere, 2022_

## Author Comment (AC1)

**Responses to the Comments from Reviewer #1**

**General Comments:**

The authors propose a combination of InSAR time-series data from one viewing direction with one ML approach (Random Forest Method) to map the permafrost deformation in the Tibetan Plateau, emphasizing the area where radar visibility problems take place. As InSAR delivers ground displacements along the slant-looking direction, visibility problems such as layover and shadow often arise in mountainous areas; it occurs in the descending track in this study. In such a case, InSAR users will usually take advantage of other data imaged from another direction that is the ascending track in this study. However, instead of using the ascending InSAR data, the authors employ the ML method to infer a permafrost stability map even at unmeasured areas. In other words, it appears as if the authors derived some signals from virtually nothing. In terms of the overall design of research work, I am not willing to recommend the authors' approach to my friends.

Response: Thank you for the reading and comments. We agree with the reviewer that the combination of ascending and descending datasets could improve the monitoring ability of ground deformation; however, in regions where the datasets are strongly affected by terrain visibility, the ground deformation could not be monitored by the combination of ascending and descending datasets. According to the previous studies (Michaelides et al., 2019; Chen et al., 2020; Li et al., 2022), the permafrost stability, which could be captured by the ground deformation rate, is closely correlated with environmental factors. Thus, the machine learning method was adopted in our study for mapping the relationship between the environmental factors and the permafrost stability. With the aid of the established mapping relationship, the permafrost stability in areas where the visibility of SAR images is poor or InSAR results are not available could be mapped. We believe that the combination of InSAR and machine learning method is a topic worthy of investigation. To avoid the confusion of the reviewer, a clarification will be added in our revision.

Chen, J., Wu, Y., O'Connor, M., Cardenas, M. B., Schaefer, K., Michaelides, R., Kling, G., 2020. Active layer freeze-thaw and water storage dynamics in permafrost environments inferred from InSAR. Remote Sensing of Environment, 248, 112007.
Li, R., Zhang, M., Konstantinov, P., Pei, W., Tregubov, O., Li, G., 2022. Permafrost degradation induced thaw settlement susceptibility research and potential risk analysis in the Qinghai-Tibet Plateau. CATENA, 214, 106239.
Michaelides, R. J., Schaefer, K., Zebker, H. A., Parsekian, A., Liu, L., Chen, J., Schaefer, S. R., 2019. Inference of the impact of wildfire on permafrost and active

layer thickness in a discontinuous permafrost region using the remotely sensed active layer thickness (ReSALT) algorithm. Environmental Research Letters, 14(3), 035007.

Furthermore, there are a couple of serious issues in the authors' interpretation of InSAR data and time-series analysis. They do not mention anything about the corrections of the tropospheric errors/artifacts. Even if they employ the time-series analysis approach with plenty of SAR images, it is impossible to ignore the tropospheric errors particularly when they use the entire image frame; the larger the imaged area, the larger the tropospheric errors. Although the authors attribute the apparent seasonal signals to the subsidence and uplift due to the freeze-thaw cycle of the active layer, we should first eliminate or minimize the tropospheric errors.

Response: Thank you for the reading and comment. Indeed, a tropospheric correction was conducted through spatial-temporal filtering (Garthwaite et al., 2013; Li et al., 2021) in our InSAR processing, even though this correction was not mentioned in our manuscript, as we believed that this kind of processing is routine in InSAR analyses. The adopted spatial-temporal filtering, which has been shown effective in minimizing the tropospheric error on the Tibetan Plateau (Garthwaite et al., 2013; Li et al., 2021), assumes that the atmospheric phase delay signal is spatially correlated but temporally uncorrelated. In addition, the verifications of the ground deformations obtained with InSAR analysis further confirmed the accuracy of our time-series InSAR processing. To avoid this confusion, a clarification will be added in our revision.

Garthwaite, M. C., Wang, H., Wright, T. J., 2013. Broadscale interseismic deformation and fault slip rates in the central Tibetan Plateau observed using InSAR. Journal of Geophysical Research: Solid Earth, 118(9), 5071-5083.
Li, R., Li, Z., Han, J., Lu, P., Qiao, G., Meng, X., Zhou, F., 2021. Monitoring surface deformation of permafrost in Wudaoliang Region, Qinghai-Tibet Plateau with ENVISAT ASAR data. International Journal of Applied Earth Observation and Geoinformation, 104, 102527.

Secondly, while it is related to the previous comment, Figure 14 derived from ascending track clearly indicates that the "deformation rates" are closely correlated with the local topography. Those are called topography (elevation)-correlated noise, which is again caused by tropospheric delays. They can be corrected, by fitting with the DEM.

Response: Thank you for the reading and comment. Indeed, the topography-correlated noise was simulated and reduced using the external DEM in our InSAR processing, and the same procedures were adopted for processing ascending and descending SAR images. To avoid this confusion, a clarification will be added in our revision.

Thirdly, while InSAR tells us the surface displacements relative to non-deformed point(s), it is not clear where the reference pixels are located; the reference pixels should be stable not only in one InSAR image but also over the entire observation period. I, therefore, recommend reanalyzing the InSAR time-series data based on the ascending track, considering the points above.

Response: Thank you for the reading and comment. In our InSAR processing, the reference pixels (i.e., Ground Control Points) were selected in the flat terrain with minimal ground deformation, and they were stable not only in one InSAR image but also over the entire observation period. To avoid this confusion, a clarification will be added in our revision.

It is not clear why they must use the Random Forest Method; permafrost stability and landslide susceptibility follow totally different physical mechanisms. The authors should show both descending and ascending data over flat areas as verification of deformation signals as they are mostly vertical.

Response: Thank you for the reading and comments. We agree with the reviewer that permafrost stability and landslide can follow different physical mechanisms; however, both permafrost stability and landslide occurrence can be correlated to environmental factors. Note that although the random forest method was adopted in our study to map the relationship between the permafrost stability and the environmental factors, other machine learning methods could also be adopted for building such a relationship. In addition, the adopted random forest method has been shown effective in mapping the permafrost degradation-induced thaw settlement susceptibility on the Tibetan Plateau (Li et al., 2022). To avoid potential confusion, more clarification will be added in our revision.

We also agree with the comment that the comparison of ground deformation over flat areas obtained from both descending and ascending data may be adopted for verifying the deformation signals. Figure R1 depicts the comparison of the ground deformation rate obtained from the ascending and descending SAR images, which confirms the accuracy of the ground deformation results obtained from our InSAR analyses. This kind of comparison will be added in our revision.

[Figure]

Figure R1. Comparision of the ground deformation rate derived from the ascending and descending observations

Li, R., Zhang, M., Konstantinov, P., Pei, W., Tregubov, O., Li, G., 2022. Permafrost degradation induced thaw settlement susceptibility research and potential risk analysis in the Qinghai-Tibet Plateau. CATENA, 214, 106239.

**Specific and technical comments:**

L30: As the focus is now on Tibet, those papers outside Tibet and/or Global should be removed.

Response: Thank you for the suggestion. References that are not related to the Tibetan Plateau will be deleted in our revision.

L48: Delete "that"

Response: Thank you for the suggestion. The revision will be made accordingly.

L51: Unclear sentence

Response: Thank you for the comment. To avoid this confusion, this sentence will be modified as follows in our revision. "The permafrost stability is often manifested by the variation of the permafrost thickness, and which could be captured by the ground deformation; thus, the permafrost stability can be captured through the monitoring of the ground deformation."

L58: The two references are not related to permafrost.

Response: Thank you for the comment. Related references will be modified.

L109: Delete "in which"

Response: Thank you for the comment. The revision will be made accordingly.

L135: Replace "spatial" with "temporal"

Response: Thank you for the comment. The revision will be made accordingly.

L176: "relatively flat and homogeneous" conflicts with L115, "mountainous terrain"

Response: Thank you for the comment. Note that the study area is relatively flat, and variations of the ground elevations in the mountainous terrain are relatively small. To avoid this confusion, related sentences will be modified in our revision.

L276: Is 0.8 true? There is a big deviation near the end.

Response: Thank you for the comment. Note that although the deviation near the end of the line was large, the value of 0.80 was correct and the overall performance of the fitting was satisfactory.

Figure 6: When are the periods in the four years? Show month and date.

Response: Thank you for the comment. The thawing periods in these four years were 1 April 2015 to 23 August 2015, 26 April 2017 to 24 August 2017, 21 April 2018 to 31 August 2018, and 28 April 2019 to 26 August 2019, respectively. This information will be provided in our revision.

L319: Disagree with "in general agreement"

Response: Thank you for the comment. Plotted in Figure 7(c) are the variations of the seasonal thaw subsidence and the ground elevation along profile AB; as can be seen, a larger elevation tends to yield smaller thaw subsidence but there are exceptions. The related expressions will be modified in our revision.

L333: Michaelides et al (2019) examined the post-fire area, where there occurred a big change in surface vegetation. But the authors are now examining unburned areas. If we follow the suggestion by Michaelides et al (2019), we expect significant deformation signals over "Bare lands" as there would be no insulation effects. The authors should check if there exist such signals.

Response: Thank you for the comment. Figures 8(b-c) depicted that a smaller NDVI value tends to result in larger thaw subsidence, and the bare lands with smaller NDVI have larger thaw subsidence; however, the thaw subsidence can be affected by various factors (e.g., elevation, slope, ice content), which caused the relationship between the thaw subsidence and the NDVI not significant. As such, the bare lands do not always yield significant deformations. To avoid potential confusion, more clarifications will be added in our revision; meanwhile, the related reference will be modified.

Figure 12: Leveling route by Wu et al (2018) should be clarified, whereas only one leveling data was shown.

Response: Thank you for the comment. In the studies of Wu et al. (2018), only one leveling site was located in the study area, thus, only one leveling data was provided in our study. More clarifications of the leveling data will be added in our revision.

Wu, Z., Zhao, L., Liu, L., Zhu, R., Gao, Z., Qiao, Y., Xie, M., 2018. Surface-deformation monitoring in the permafrost regions over the Tibetan Plateau, using Sentinel-1 data. Sci. Cold Arid Reg. 10(2), 114-125.

---

## Author Comment (AC2)

**Responses to the Comments from Reviewer #2**

The paper aims to develop a method for permafrost stability mapping on the Tibetan Plateau, which integrates InSAR and random forest. The work is an innovative and very worthwhile attempt, and it has a good guiding for disaster research in some regions with a complex geological environment like the Qinghai Tibet Plateau particularly. However, some minor issues still need to be improved. The specific comments are given as follows.

Response: Thank you for the careful reading and kind words. We sincerely appreciate the comments that have helped sharpen this paper. Specific responses to the review comments are presented immediately after the respective review comments.

1. There are two spelling mistakes in line 48 and 305 that "too that many" and a sudden "s".

Response: Thank you for the careful reading and comment. The spelling mistakes will be modified in our revision.

2. Line 177: please explain why do you use vertical ground deformation, but not LOS ground deformation, i.e. what are the advantages over here by doing so?

Response: Thank you for the careful reading and comment. Note that the main ground deformation in permafrost areas is the thaw subsidence or frost heave, which can be manifested in the vertical ground deformation. Thus, the vertical ground deformation, rather than the LOS deformation, was adopted for analyzing the permafrost stability. According to the suggestion, more clarifications will be added in our revision.

3. It is mentioned in line 265 that permafrost instability mainly distributed in the valley areas with low altitude. However, in your Fig. 4(a), there are many areas with high deformation that distribute in high altitude mountainous areas. Please explain!

Response: Thank you for the careful reading and comment. The ground deformation zones are mainly concentrated in the valley areas with low altitudes, where the water content is relatively high. However, permafrost stability can be affected by various environmental factors. For example, the land cover type in some high-altitude mountainous areas is the bare lands with no vegetation coverage, which are also

susceptible to ice melting and thaw subsidence. To avoid this confusion, more clarifications will be provided in our revision.

4. It is mentioned in line 266 that the ground deformation mainly took place in the west-facing slopes. In theory, it is right due to the "descending" approach of the satellite. However, in Figure 4b and 4c it seems that there are more points on the east-facing slopes. Why?

Response: Thank you for the careful reading and comment. The terrain visibility of the descending SAR images in east-facing slopes is mainly foreshortening, which causes the ground deformation results obtained in east-facing slopes not reliable. As such, although there are lots of deformation points located on east-facing slopes, the deformation results are not reliable, which could not be adopted to indicate permafrost degradation. More deformation points in Figure 4(b), compared to Figure 4(c), might be attributed to the higher coherence of the interferograms. To avoid this confusion, more clarifications will be provided in our revision.

5. In line 370, the threshold values are set as ±0.15 mm/year and -40 mm/year. Please state or provide a scientific basis of setting up such values.

Response: Thank you for the careful reading and comment. According to the Google Earth images, the permafrost instability areas with obvious unstable characteristics (e.g., retrogressive thaw slumps and failed slopes) are usually located in the areas with a ground deformation rate smaller than -40 mm/year. Thus, in this study, the ground point with a ground deformation rate smaller than -40 mm/year and obvious unstable characteristics is classified as an unstable ground point.

Further, the stable ground points are determined according to the ground deformation rate and the image characteristics. In general, the area with a ground deformation rate close to 0 mm/year could be classified as a stable area, thus the threshold value of the ground deformation rate for stable ground should be set at a value close to 0 mm/year; and, an equal number of stable ground points should be identified in the high-quality area to avoid the potential bias in the selection of samples. Based on these two reasons, the threshold value of the ground deformation rate for stable ground was set at ±0.15 mm/year. Thus, the ground point with a deformation rate ranging from -0.15 mm/year to 0.15 mm/year and no obvious unstable characteristics is classified as a stable point.

In the potential revision, more clarifications of the criteria adopted for determining the stable and unstable ground points will be added in our revision to avoid confusions.

6. In line 397, the ROC curve is used to evaluate the accuracy of the model, but where

is the ROC figure?

Response: Thank you for the careful reading and comment. The ROC curve shown in Figure R1 will be added in our revision.

[Figure]

Figure R1. Validation of the trained random forest model using ROC curve

---

## Author Comment (AC3)

**Responses to the Comments from Reviewer #3**

This study proposes a new method that integrates InSAR time series and machine learning (random forest) for mapping permafrost stability in a selected region in central Tibet. This is probably one of the first efforts of such integration tailored towards estimating stability. However, the current framework relating InSAR-estimated surface deformation to permafrost stability is both conceptually flawed and poorly explained. The quality of the manuscript in its current form is further jeopardized by vague and even sometimes careless description and presentation. I raise a few major issues as detailed below and choose not to list minor editorial comments.

Response: Thank you for your careful reading and comments. We sincerely appreciate the comments that will help sharpen this paper. In the revision, the methodology will be made clearer and the presentation will be polished further. Further, the concept relating InSAR-estimated surface deformation to permafrost stability will be clarified. And, specific responses to the review comments are presented immediately after the respective review comments.

1. To permafrost scientists and engineers, 'permafrost stability' can be expressed from various perspectives, e.g., mechanical strength, permafrost thickness, temperature, active layer thickness, ground ice content, thaw settlement, engineering, hydrology, and even carbon stock and fluxes. In the abstract and introduction, the authors put forward a link between ground deformation (or should be ground *surface* deformation) as an observable for permafrost stability and the whole study has built on this concept. Since the authors didn't define what they mean by 'permafrost stability', I had to guess they were mainly concerned with thaw settlement.

Response: Thank you for the comments. We agree with the reviewer that permafrost stability can be expressed from various perspectives; however, the permafrost stability is often manifested by the variation of the permafrost thickness and that of the active layer thickness, and the variation of the permafrost thickness could be captured by the ground deformation rate. Meanwhile, the permafrost stability can be influenced by the soil water content, which is closely related to the seasonal thaw subsidence (Chen et al., 2020). Figure 7 and Figure 8 depict that the topography and vegetation coverage exhibit great influences on the seasonal thaw subsidence; as such, the environmental factors also have impacts on the permafrost stability and related ground deformation rate. For simplicity, the ground deformation rate was adopted as an important index for assessing the stability and degradation of the permafrost in this study. To avoid

potential confusion, more clarifications of the permafrost stability will be added in our revision.

Chen, J., Wu, Y., O'Connor, M., Cardenas, M. B., Schaefer, K., Michaelides, R., Kling, G. 2020. Active layer freeze-thaw and water storage dynamics in permafrost environments inferred from InSAR. Remote Sensing of Environment, 248, 112007.

2. There are a few fundamental and practical issues that need to be critically addressed if using thaw settlements to infer permafrost stability.

(a) As pointed out by the authors, thaw-season subsidence is predominately caused by thawing of the active layer, not permafrost. The authors used an annual cycle + linear trend to separate seasonal and linear deformation (eq 1). Such a simple model is a reasonable choice and was adopted in many InSAR studies on permafrost. However, it is clear from Figure 5 (esp. P2) that InSAR time series often deviate from this simple time pattern. Such deviations may cause errors in the estimated trend, esp. given that the duration of the InSAR time series is only about 6 years.

Response: Thank you for the comments. In fact, the seasonal thaw subsidence is often induced by the thawing of the active layer, while the long-term ground subsidence (or overall trend) could be induced by the thawed permafrost. It should be noted that the long-term ground subsidence induced by thawed permafrost could lead to permafrost instability or degradation. In this study, an empirical model was built to capture the overall trend of the ground surface deformation. Although there occurred deviations in the estimated deformation trend, and the errors caused by such deviations might come from external interferences such as human disturbance, such deviations cannot affect much the overall trend of the ground deformations, e.g., the R-squares of these three ground points were high (i.e., 0.50, 0.80, and 0.83), indicating that the time-series ground deformations in the study area could be well captured by the empirical models, as depicted in Figure 5(a). In addition, such an empirical model has been excessively adopted for the decomposition of ground deformations in permafrost areas (Colesanti et al., 2003; Lu et al., 2019). Moreover, it should be informed that the decomposition of the ground deformation using the empirical model was mainly conducted to verify the accuracy of InSAR analysis results in permafrost areas and determine the periods of thawing and frozen seasons; and, the input to the built random forest model was the ground deformation rate estimated from time-series InSAR analyses. In other words, deviations in the empirical models shown in Figure 5(a) would not affect the accuracy of the permafrost stability mapping in the study area. To avoid the confusions, more clarifications of the empirical models in Figure 5(a) will be added in our revision.

Colesanti, C., Ferretti, A., Novali, F., Prati, C., Rocca, F., 2003. SAR monitoring of progressive and seasonal ground deformation using the permanent scatterers technique. IEEE Transactions on Geoscience and Remote Sensing, 41(7), 1685-1701.

Lu, P., Han, J., Hao, T., Li, R., Qiao, G., 2020. Seasonal deformation of permafrost in Wudaoliang basin in Qinghai-Tibet plateau revealed by StaMPS-InSAR. Marine Geodesy, 43(3), 248-268.

(b) Since the presented results include both seasonal and linear deformation and in many places the authors didn't explicitly state whether the deformation is seasonal or trend, I got completely lost and wasn't sure what kinds of deformation are presented and what were used as input into the random forest. E.g. it is unclear what kind(s) of deformation are shown in Figures 7 and 8. I can only guess from the units that they are thaw-season subsidence. But wouldn't you mainly use the trend?

Response: Thank you for the comments. It should be informed that the decomposition of the ground deformation using the empirical model was mainly conducted to verify the accuracy of InSAR analysis results in permafrost areas and determine the periods of thawing and frozen seasons; and, the input to the built random forest model was the ground deformation rate estimated from time-series InSAR analyses.

Figures 7 and 8 illustrate the influences of the topography and vegetation coverage on the seasonal thaw subsidence. Note that the permafrost stability can be influenced by the soil water content, which is closely related to the seasonal thaw subsidence. The main purpose of Figures 7 and 8 was to depict that the permafrost stability can be affected by the environmental factors of topography and vegetation coverage.

In the potential revision, more clarifications of the ground deformations mentioned in the main text will be added in our revision to avoid confusions.

(c) Only till the result section 4.2.1, the authors stated the thresholds in deformation trends to classify stable vs unstable ground as "+/- 0.15 mm/year and -40 mm/year". Such important criteria need to be justified and introduced in the methodology. What are the bases of these thresholds? E.g., why trends larger than 40 mm/yr mean unstable permafrost, as it may seem small to different experts. Are these trends in the vertical direction or in line-of-sight (LOS) direction? 0.15 mm/year is extremely small compared to nominal uncertainties of InSAR measurements. Without estimating the uncertainties in the measured trend, is it still meaningful to set such a small threshold?

Response: Thank you for the comments. Note that the main ground deformation in permafrost areas can be the thaw subsidence and frost heave, which can be manifested in the vertical ground deformation. Thus, the vertical ground deformation, rather than the LOS deformation, was adopted in this study for analyzing the permafrost stability.

According to the Google Earth images, the permafrost instability areas with obvious unstable characteristics (e.g., retrogressive thaw slumps and failed slopes) are usually

located in the areas with a ground deformation rate smaller than -40 mm/year. Thus, in this study, the ground point with a deformation rate smaller than -40 mm/year and obvious unstable characteristics is classified as an unstable ground point. In reference to Zhang et al. (2020), the maximum subsidence rate of the permafrost instability area that is located in the central Tibetan Plateau was about -30 mm/year. In other words, the threshold value of -40 mm/year adopted in this study is relatively conservative. In fact, apart from the threshold value of the ground deformation rate (i.e., -40 mm/year), the unstable ground points determined were verified with the Google Earth images.

The stable ground points were also determined according to the ground deformation rate and the image characteristics. In general, the area with a ground deformation rate close to 0 mm/year could be classified as a stable area, thus the threshold value of the ground deformation rate for stable ground should be set at a value close to 0 mm/year. And, an equal number of stable ground points should be identified in the high-quality area to avoid the potential bias in the selection of samples. Based on these two reasons, the threshold value of the ground deformation rate for stable ground was set at ±0.15 mm/year. Thus, the ground point with a deformation rate ranging from -0.15 mm/year to 0.15 mm/year and no obvious unstable characteristics is classified as a stable point.

It must be noted that the decomposition of the ground deformation using the empirical model was mainly conducted to verify the accuracy of InSAR analysis results in permafrost areas and determine the periods of thawing and frozen seasons; and, the input to the random forest model was the ground deformation rate estimated from time-series InSAR analyses. As such, the deviations or uncertainties in the empirical models shown in Figure 5(a) would not affect the accuracy of the permafrost stability mapping in the study area.

In the potential revision, more clarifications of the criteria adopted for determining the stable and unstable ground points will be added in our revision to avoid confusions; and, more clarifications of the ground deformations mentioned in the main text will be added.

Zhang, X., Zhang, H., Wang, C., Tang, Y., Zhang, B., Wu, F., Zhang, Z., 2020. Active Layer Thickness Retrieval Over the Qinghai-Tibet Plateau Using Sentinel-1 Multitemporal InSAR Monitored Permafrost Subsidence and Temporal-Spatial Multilayer Soil Moisture Data. IEEE Access, 8, 84336-84351.

(d) Not stated explicitly in the paper, but I suppose the authors converted LOS deformation trends to vertical by assuming the ground motion is purely vertical; and used vertical trends as input to random forest and all the InSAR results presented are in the vertical direction. Then another major flaw lies in the ignorance of lateral flow on slopes in periglacial landscapes. Depending on the geomorphic type and nature of the processes, lateral movement on certain such as landforms such as solifluction

sheets, rock glaciers, and even fluvial fans can move (much) faster than 40 mm/year, yet the underlying permafrost could be stable. Whereas mass wasting associated with thermokarst processes such as active layer detachment slides and thaw slumps (Figure 16 shows one example) can show very fast movement due to degrading permafrost. Without differencing the nature of deformation on flat vs slope regions and knowing the surface geomorphology, the inferred 'permafrost stability' would be unreliable over slopes.

Response: Thank you for the comments. Note that the study area is a relatively flat and homogeneous area, and no active fault is developed, thus, the ground deformation in the study area is assumed to be concentrated in the vertical direction. Further, the major ground deformation in permafrost areas could be the thaw subsidence and frost heave, which can be manifested in the vertical ground deformation. Thus, the vertical ground deformation, rather than the LOS deformation, was employed in this study for analyzing the permafrost stability. In this study, the LOS deformation was converted to vertical ground deformation based on the incidence angle of satellite LOS. In fact, the kind of ground deformation transformation is reliable and which has been widely adopted (Liu et al., 2010; Chang and Hanssen, 2015; Lu et al., 2020); thus, we believe our analysis results will not be affected much by this transformation.

It is noted that although the mass wasting associated with thermokarst processes, such as active layer detachment slides and thaw slumps depicted in Figure 16(c), may show fast movement due to degrading permafrost, the long-term ground deformations prior to slope failures could be small, and which can be detected by the InSAR method. As a matter of fact, apart from the threshold value of the ground deformation rate, the unstable ground points determined in this study were further verified with the Google Earth images. Thus, the ground points with large ground deformation rates induced by other reasons can be excluded. In addition, the glacier movement or slope deformation could reach several meters per year, and such large ground deformations could not be detected by the time-series InSAR method due to the decoherence.

In the potential revision, more clarifications of the criteria adopted for determining the stable and unstable ground points will be added in our revision to avoid confusions.

Liu, L., Zhang, T., Wahr, J., 2010. InSAR measurements of surface deformation over permafrost on the North Slope of Alaska. Journal of Geophysical Research: Earth Surface, 115(F3).
Chang, L., Hanssen, R. F., 2015. Detection of permafrost sensitivity of the Qinghai–Tibet railway using satellite radar interferometry. International journal of remote sensing, 36(3), 691-700.
Lu, P., Han, J., Hao, T., Li, R., Qiao, G., 2020. Seasonal deformation of permafrost in Wudaoliang basin in Qinghai-Tibet plateau revealed by StaMPS-InSAR. Marine Geodesy, 43(3), 248-268.

(e) Yet, one of the selling points of this work is to use random forest to fill gaps in 'poor visibility areas' on slopes. This machine-learning-enabled advantage cannot solve the fundamental issue raised in (d).

Response: Thank you for the comments. We agree with the reviewer that the novelty of this study was the permafrost stability mapping integrating the time-series InSAR and random forest method, thus, the permafrost stability in areas where the visibility of SAR images is poor or InSAR analysis results are not available could be mapped.

As mentioned above, apart from the threshold value of the ground deformation rate, the unstable ground points determined in this study were also verified with the Google Earth images. Thus, the ground points with large ground deformation rates induced by other reasons can be excluded; and, the accuracy of the database adopted for training the random forest model could be guaranteed. Further, the glacier movement or slope deformation could reach several meters per year, and such large ground deformations cannot be detected by the time-series InSAR method due to the decoherence.

In the potential revision, more clarifications of the criteria adopted for determining the stable and unstable ground points will be added in our revision to avoid confusions; and, the novelty of this study will be further highlighted.

3. The methodologic description of how integrating InSAR with random forest to infer permafrost stability is very vague to me. I raised a few concerns related to this methodology above and would summarize the key ones below.

Response: Thank you for the comments. In this study, the ground deformation rate estimated from the time-series InSAR analysis was adopted to indicate the permafrost stability; however, the main problem of such analysis is that the permafrost stability could not be well mapped in areas where the visibility of SAR images is poor or the InSAR analysis results are not available. It must be noted that the permafrost stability is correlated with the environmental factors (e.g., elevation, aspect, slope, curvature, land cover, NDVI, land surface temperature, and distance to the Qinghai-Tibet Highway), thus, the mapping relationship between the permafrost stability and the environmental factors was first established through a random forest model; then, the permafrost stability in areas where the visibility of SAR images is poor or the InSAR analysis results are not available was mapped with the trained random forest model.

To avoid this confusion, more detailed information of this integrated method will be added in our revision.

(a) InSAR observations: seasonal or trend or both? No uncertainties.

Response: Thank you for the comments. In this study, the ground deformation rate estimated from the time-series InSAR analysis, rather than the long-term deformation trend, was adopted to indicate the permafrost stability; whereas, the seasonal thaw subsidence estimated from the time-series InSAR analysis was utilized to analyze the influences of topography and vegetation coverage on the permafrost stability.

Note that the decomposition of the ground deformation using the empirical model was mainly conducted to verify the accuracy of InSAR analysis results in permafrost areas and determine the periods of thawing and frozen seasons; and, the input to the random forest model was the ground deformation rate estimated directly from the time-series InSAR analysis, thus, the deviations or uncertainties in the empirical models shown in Figure 5(a) would not affect the accuracy of the permafrost stability mapping in the study area; and, the uncertainties of the InSAR observations trend were not analyzed in this study.

In the potential revision, more clarifications of the ground deformations mentioned in the main text will be added in our revision to avoid confusions.

(b) Classification of stable vs unstable: what are the bases?

Response: Thank you for the comments. According to the Google Earth images, the permafrost instability areas with obvious unstable characteristics (e.g., retrogressive thaw slumps and failed slopes) are usually located in areas with a ground deformation rate smaller than -40 mm/year. Thus in this study, the ground point with a deformation rate smaller than -40 mm/year and obvious unstable characteristics was classified as an unstable ground point.

The stable ground points were also determined according to the ground deformation rate and the image characteristics. In general, the area with a ground deformation rate close to 0 mm/year could be classified as a stable area, thus the threshold value of the ground deformation rate for stable ground should be set at a value close to 0 mm/year. And, an equal number of stable ground points should be identified in the high-quality area to avoid the potential bias in the selection of samples. Based on these two reasons, the threshold value of the ground deformation rate for stable ground was set at $\pm0.15$ mm/year. Thus, the ground point with a deformation rate ranging from -0.15 mm/year to 0.15 mm/year and no obvious unstable characteristics was classified as stable point.

In the potential revision, more clarifications of the criteria adopted for determining the stable and unstable ground points will be added in our revision to avoid confusions.

(c) What are the exact inputs and outputs of the random forest?

Response: Thank you for the comments. The data adopted for training the random forest model were the stable and unstable ground points determined with the criteria mentioned above in the study area. Note that the inputs to the random forest model were the environmental factors such as the ground elevation, aspect, slope, curvature, land cover, NDVI, land surface temperature, and distance to Qinghai-Tibet Highway, whereas the outputs of the random forest model were the permafrost stability mapping results in the study area. To avoid this confusion, more clarifications will be added in our revision.

(d) What are the reasons for selecting the topographic and climate variable? E.g. it makes little intuitive sense to include curvature and it turns out that curvature is the least important factor.

Response: Thank you for the comments. The reasons for selecting the topographic and climate variables as the environmental factors could be summarized as follows:

1) The topographic variables (i.e., ground elevation, aspect, slope, and curvature) can exhibit impacts on the permafrost stability through altering the soil water content and affecting vegetation coverage; and, the climate variable (i.e., land surface temperature) might also influence the vegetation coverage and soil water content, which could then exhibit impacts on the permafrost stability.

2) The random forest model trained in this study indicated that the slope and aspect yield the most significant impact on the permafrost stability, then is the distance to the Qing-Tibet Highway, land surface temperature, NDVI, elevation, land cover, and curvature. Note that although curvature yields the least impact on permafrost stability, it could not be ignored in the permafrost stability mapping.

3) In reference to Deluigi et al. (2017) and Chen et al. (2020), the permafrost stability can be affected by the topography factors; and, the studies in Qin et al. (2020) depict that the land surface temperature is a good indicator for analyzing permafrost stability.

To avoid this confusion, more clarifications on the selection of the topographic and climate variables as environmental factors will be added in our revision.

Chen, J., Wu, Y., O'Connor, M., Cardenas, M. B., Schaefer, K., Michaelides, R., Kling, G., 2020. Active layer freeze-thaw and water storage dynamics in permafrost environments inferred from InSAR. Remote Sensing of Environment, 248, 112007.
Deluigi, N., Lambiel, C., Kanevski, M., 2017. Data-driven mapping of the potential mountain permafrost distribution. Science of The Total Environment, 590, 370-380.
Qin, Y., Zhang, P., Liu, W., Guo, Z., Xue, S., 2020. The application of elevation corrected MERRA2 reanalysis ground surface temperature in a permafrost model on

the Qinghai-Tibet Plateau. Cold Regions Science and Technology, 175, 103067.

(e) What land surface temperature did you use? Annual ground surface temperature?

Response: Thank you for the comments. The land surface temperature adopted in this study was the annual average land surface temperature from 2014 to 2020. To avoid this confusion, more clarifications will be added in our revision.

The authors should pay more attention to properly citing references and following scientific rigor. Here are a few examples:

Response: Thank you for the careful reading and comment. The references cited in this paper will be carefully checked, and unsuitable references will be avoided in the revision.

Line 40: Schaefer et al., 2015 was about retrieving active layer thickness from InSAR, not about using the thickness as an index for permafrost stability.

Response: Thank you for the careful reading and comment. The unsuitable reference will be replaced by more suitable references in the revision.

Line 44-45: The sentence is about sparse field-based measurements. But the two papers cited are both based on remote sensing.

Response: Thank you for the careful reading and comment. The unsuitable references will be replaced by more suitable references in the revision.

Line 48-49: Widhalm et al., 2017 mainly used SAR data, didn't involve 'many environmental factors'.

Response: Thank you for the careful reading and comment. The unsuitable reference will be replaced by more suitable references in the revision.

Line 62: Schaefer et al., 2015 was not a Tibet study.

Response: Thank you for the careful reading and comment. The unsuitable reference will be replaced by more suitable references in the revision.

Line 140: Ran et al. (2021) were concerned with permafrost temperature and thickness, not ground deformation.
(This list can be very long)
Two relevant and important papers were published recently and should provide some guidance and inspiration.

Ran et al. "Biophysical permafrost map indicates ecosystem processes dominate permafrost stability in the Northern Hemisphere." Environmental Research Letters 16.9 (2021): 095010.
Chen et al. "Magnitudes and patterns of large-scale permafrost ground deformation revealed by Sentinel-1 InSAR on the central Qinghai-Tibet Plateau." Remote Sensing of Environment 268 (2022): 112778.

Response: Thank you for the careful reading and comment. The unsuitable reference will be replaced by the suggested references in the revision.

In addition to the issues raised earlier, there are a few places of superficial or incorrect expressions of permafrost concept, such as:

Response: Thank you for the careful reading. The unsuitable expressions in this paper will be replaced by more suitable expressions in the revision.

Line 28: shrinking of 'permafrost boundaries' should be 'permafrost extent.

Response: Thank you for the comment. The revision will be made accordingly.

Line 34: thickening active layer is not the root cause of carbon release from permafrost.

Response: Thank you for the careful reading. We agree with the reviewer that increase of the active layer thickness is not the root cause of carbon release from permafrost, and the carbon is mainly released because of the permafrost warming and degradation. The related sentence will be modified in the revision to avoid this confusion.

Line 278: ground surface, not permafrost, heaves; ground (the active layer to be exact) is freezing, not frozen in September.

Response: Thank you for the careful reading. The related sentence will be modified according to this suggestion in the revision.

Line 519: deterioration should be degradation.

Response: Thank you for the comment. The revision will be made accordingly.